# The Gram-positive bacterium *Romboutsia ilealis* harbors a polysaccharide synthase that can produce (1,3;1,4)-β-D-glucans

Shu-Chieh Chang[1,2], Mu-Rong Kao [1,2], Rebecka Karmakar Saldivar[1,2], Sara M. Díaz-Moreno [1], Xiaohui Xing [3], Valentina Furlanetto[4], Johannes Yayo[4], Christina Divne [4], Francisco Vilaplana [1], D. Wade Abbott[3] & Yves S. Y. Hsieh [1,2] ✉

(1,3;1,4)-β-D-Glucans are widely distributed in the cell walls of grasses (family Poaceae) and closely related families, as well as some other vascular plants. Additionally, they have been found in other organisms, including fungi, lichens, brown algae, charophycean green algae, and the bacterium *Sinorhizobium meliloti*. Only three members of the *Cellulose Synthase-Like* (*CSL*) genes in the families *CSLF*, *CSLH*, and *CSLJ* are implicated in (1,3;1,4)-β-D-glucan biosynthesis in grasses. Little is known about the enzymes responsible for synthesizing (1,3;1,4)-β-D-glucans outside the grasses. In the present study, we report the presence of (1,3;1,4)-β-D-glucans in the exopolysaccharides of the Gram-positive bacterium *Romboutsia ilealis* CRIB[T]. We also report that *RiGT2* is the candidate gene of *R. ilealis* that encodes (1,3;1,4)-β-D-glucan synthase. *Ri*GT2 has conserved glycosyltransferase family 2 (GT2) motifs, including D, D, D, QXXRW, and a *C*-terminal PilZ domain that resembles the *C*-terminal domain of bacteria cellulose synthase, BcsA. Using a direct gain-of-function approach, we insert *RiGT2* into *Saccharomyces cerevisiae*, and (1,3;1,4)-β-D-glucans are produced with structures similar to those of the (1,3;1,4)-β-D-glucans of the lichen *Cetraria islandica*. Phylogenetic analysis reveals that putative (1,3;1,4)-β-D-glucan synthase candidate genes in several other bacterial species support the finding of (1,3;1,4)-β-D-glucans in these species.

(1,3;1,4)-β-D-Glucans are unbranched homoglucans composed of both 3- and 4-linked β-glucopyranosyl (Glc*p*) residues. The occurrence of the (1,3;1,4)-β-D-glucans has been the subject of considerable interest[1,2]. They are best known in the cell walls of cereals and other grasses (family Poaceae). However, they have also been found in some other vascular plants, including those in families closely related to the Poaceae, the lycophyte *Selaginella moellendorffii*, and in horsetails

(*Equisetum* spp.), as well as other monilophytes (ferns)[3]. Additionally, they have been occasionally reported in several other taxa, including charophycean green algae[4], brown algae[5], fungi[6], and the fungal symbiont of lichens, such as *Cetraria islandica* (where the (1,3;1,4)-β-D-glucan is known as lichenin or lichenan)[7], as well as the Gram-negative bacterium *Sinorhizobium meliloti*[8]. In the backbone of (1,3;1,4)-β-D-glucans, two and three consecutive (1→4) bonds [β-Glc*p*-(1→4)-β-

[1]Division of Glycoscience, Department of Chemistry, School of Engineering Sciences in Chemistry, Biotechnology and Health, KTH Royal Institute of Technology, AlbaNova University Centre, Stockholm SE10691, Sweden. [2]School of Pharmacy, College of Pharmacy, Taipei Medical University, 250 Wuxing Street, Taipei 11031, Taiwan. [3]Lethbridge Research and Development Centre, Agriculture and Agri-Food Canada, Lethbridge, AB T1J 4B1, Canada. [4]Department of Industrial Biotechnology, School of Engineering Sciences in Chemistry, Biotechnology and Health, KTH Royal Institute of Technology, AlbaNova University Centre, Stockholm SE10691, Sweden. ✉e-mail: yvhsieh@kth.se

Glc*p*-(1 → 4)-β-Glc*p* (a cellotriosyl unit) and β-Glc*p*-(1 → 4)-β-Glc*p*-(1 → 4)-β-Glc*p*-(1 → 4)-β-Glc*p* (a cellotetraosyl unit), respectively] are usually found separated by a single (1 → 3) bond; consecutive (1 → 3) bonds have never been found. The proportions of the cellotriosyl and cellotetraosyl units can be determined by treating the (1,3;1,4)-β-D-glucans with the enzyme lichenase (EC 3.2.1.73), which hydrolyzes them by breaking all the (1 → 4) bonds that immediately follow (1 → 3) bonds on the reducing end side, producing the triose β-Glc*p*-(1 → 4)-β-Glc*p*-(1 → 3)-β-Glc (abbreviated to G4G3G, DP3) and the tetraose β-Glc*p*-(1 → 4)-β-Glc*p*-(1 → 4)-β-Glc*p*-(1 → 3)-β-Glc (G4G4G3G, DP4) oligosaccharides that are easily separated and quantified. The molar ratio of these two oligosaccharides (DP3/DP4 ratio) varies widely with taxa, indicating variation in the molecular assembly of (1,3;1,4)-β-D-glucans[1]. (1,3;1,4)-β-D-glucans, with very high or very low DP3/DP4 ratios, such as lichenin that has a very high ratio of 20.2–24.6:1, are more regular and hence can more readily align over extended regions, resulting in them being insoluble[7,9]. (1,3;1,4)-β-D-glucans from cereals and other grasses have intermediate DP3/DP4 ratios of 2–4.5:1, suggesting these (1,3;1,4)-β-D-glucans are irregular and soluble in the matrix phase of primary cell walls of the Poaceae[1,7,9].

Different genes and enzymes are involved in the biosynthesis of (1,3;1,4)-β-D-glucans, depending on the organisms involved. In the Poaceae, the (1,3;1,4)-β-D-glucans are synthesized by Carbohydrate Active enZyme (CAZy) family 2 glycosyltransferases (GT2) encoded by *cellulose-synthase-like* (*CSL*) gene families. To date, only the *CSLF*, *CSLH*, and *CSLJ* gene families have been characterized and are restricted to the Poaceae and other monocotyledons[10–12]. In the lycophyte *Selaginella moellendorffii*, (1,3;1,4)-β-D-glucan may be produced by a synthase encoded by an ortholog of the *AGlcS* gene that in the moss *Physcomitrella patens* encodes a GT2 enzyme that produces a mixed-linkage (1,3;1,4)-arabinoglucans[13]. Interestingly, these *S. moellendorffii* and *P. patens* GT2 enzymes resemble a fungal (1,3;1,4)-β-D-glucan synthase: the Three-Four Transferase 1 (Tft1) of *Aspergillus fumigatus*[6]. In brown algae, which have only cellotriosyl units separated by single (1 → 3) bonds in their (1,3;1,4)-β-D-glucans, their putative (1,3;1,4)-β-D-glucan synthase genes are genetically more closely related to the *Bacterial cellulose synthase* (*Bcs*) gene[5]. The enzymes responsible for the synthesis of bacterial exopolysaccharides (EPSs) are often encoded by the genes in the bacterial GT2 family, for example, the *Bacterial cellulose synthase A* (*BcsA*) and *Curdlan synthase* (*CrdS*)[14]. The structural composition of EPSs has a strong correlation with bacterial pathogenicity, including bacteria–host interaction, resistance to host immune response, and the formation of biofilms[15]. In the bacterium *S. meliloti*, the (1,3;1,4)-β-D-glucans in the EPS are unusual in having only cellobiosyl units [β-Glc*p*-(1 → 4)-β-Glc*p*] separated by single (1 → 3) bonds and are implicated in bacterial aggregation and biofilm formation[8], as well as in host plant root attachment. Of the putative bacterial genes currently identified in (1,3;1,4)-β-D-glucan biosynthesis direct gain-of-function has not been demonstrated. The *Bcs* operon encodes for a protein, which contains D, D, D, QXXRW, and PilZ domains, responsible for the biosynthesis of bacterial cellulose[16,17]. It has therefore been suggested that the two genes operon *bgsBA* encoding protein (with "*bgs*" standing for "β-glucan synthesis"), which also contains D, D, D, QXXRW motifs, could be involved in the (1,3;1,4)-β-D-glucan biosynthesis in *S. meliloti*[8]. Even though BgsA does not have a PilZ domain, the expressed BgsA *C*-terminal peptide can still bind to the bacterial secondary messenger cyclic diguanylate (c-di-GMP)[8], which is consistent with the finding of *Cereibacter sphaeroides* cellulose synthase (BcsA), in which activity is facilitated through c-di-GMP binding[18–22].

Very few bacterial GTs have had their function biochemically characterized and, of these, most were from Gram-negative bacteria. The small intestine microbes are typically Gram-positive bacteria[23], and their EPSs have a major influence on the microbiota population[24], as well as helping commensals to prevent pathogen colonization[25]. Probiotic microorganisms contribute to the modulation of intestinal commensals and benefits in health. To this end, a commensal bacterium in a novel phylotype, named CRIB, was found to be increased by probiotic intake, which resulted in a decrease in the severity of pancreatitis and associated sepsis in an experimental rat model for acute pancreatitis studies[26]. *Romboutsia ilealis* CRIB[T] was then described as the type strain of the genus of *Romboutsia*. The genus *Romboutsia* is in the family Peptostreptococcaceae which has several members that are common intestinal microbes, including *Clostridioides difficile*[27]. Most *Romboutsia* species have an intestinal origin, but their roles in the gastrointestinal tract of humans and mammals are less well characterized.

Here we report the presence of (1,3;1,4)-β-D-glucans in the EPS of the Gram-positive bacterium *R. ilealis* CRIB[T] and investigate the role of the *RiGT2* gene in (1,3;1,4)-β-D-glucan biosynthesis using a stably transfected *Saccharomyces cerevisiae* yeast model (Fig. 1). Our findings demonstrate the involvement of a prokaryotic GT2 in the synthesis of (1,3;1,4)-β-D-glucans. The heterologously synthesized (1,3;1,4)-β-D-glucans exhibit a high proportion of cellotriosyl units, similar to that found in lichenin[7,28].

## Results

### (1,3;1,4)-β-D-glucan is a component of *R. ilealis* exopolysaccharides

We prepared *R. ilealis* CRIB[T] EPSs, which were chemically converted to partially methylated alditol acetates (PMAAs) mixture for glycosidic

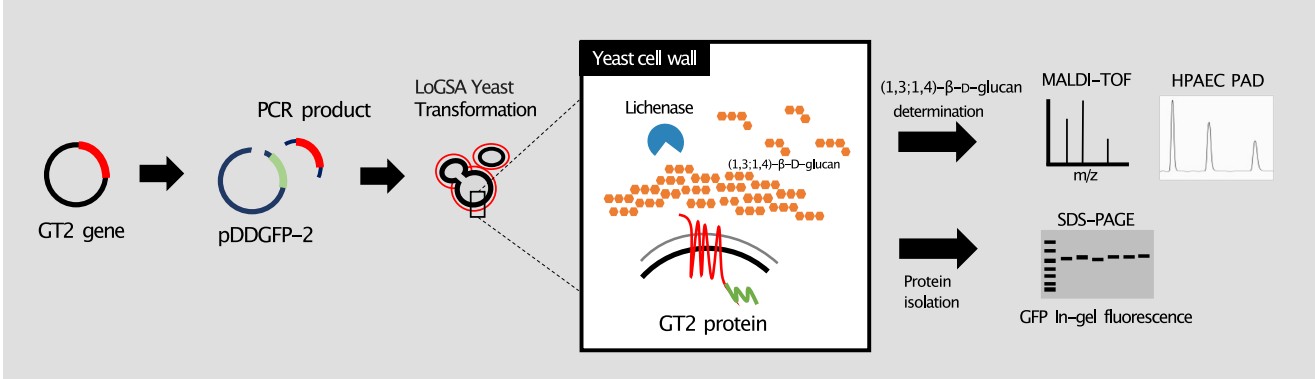

**Fig. 1 | Exploring the functions of the *GT2* polysaccharide synthase gene through gain-of-function experiments.** The cDNA encoding GT2 polysaccharide synthase was cloned into the pDDGFP-2 vector through homologous recombination. The resulting construct was then transformed into the quadruple knockout yeast *Saccharomyces cerevisiae* LoGSA[33]. Expression of the *GT2* gene was carried out following the methodology described in Newstead et al.[32] and Drew et al.[31]. Post-transformation, positive expression clones were identified through in-gel fluorescence screening. The polysaccharide synthesized by the overexpressed GT2 proteins was subsequently digested by associated polysaccharide hydrolases. The resulting oligosaccharide profiles were analyzed by MALDI-TOF MS and HPAEC-PAD.

linkage analyses (Supplementary Fig. 1). All EPS preparations had 4-linked and 3-linked glucopyranosyl residues in relatively high abundance. The 4-linked Glc*p*: 3-linked Glc*p* ratio was approximately 4:1, which is higher than that of grass (1,3;1,4)-β-D-glucans (2.2:1–2.6:1)[29].

EPS preparations were also treated with lichenase, and the enzyme-released oligosaccharides were separated by high-performance anion-exchange chromatography with pulsed amperometric detection (HPAEC-PAD). Retention times of the analytes were compared with those of a series of commercially available (1,3;1,4)-β-D-glucan oligosaccharides of known structures obtained by treating barley (1,3;1,4)-β-D-glucans with lichenase (Fig. 2). Although several unexpected oligosaccharides were detected in both the control (buffer only) and the lichenase-treated samples, two oligosaccharides released from *R. ilealis* CRIB[T] EPSs eluted at the same retention times as the commercially available DP3 (G4G3G$_R$) and DP4 (G4G4G3G$_R$), derived from barley (1,3;1,4)-β-D-glucan. The ratio of DP3 to DP4 was close to 2:1. This result confirmed that (1,3;1,4)-β-D-glucans are secreted by *R. ilealis* CRIB[T] in its EPSs.

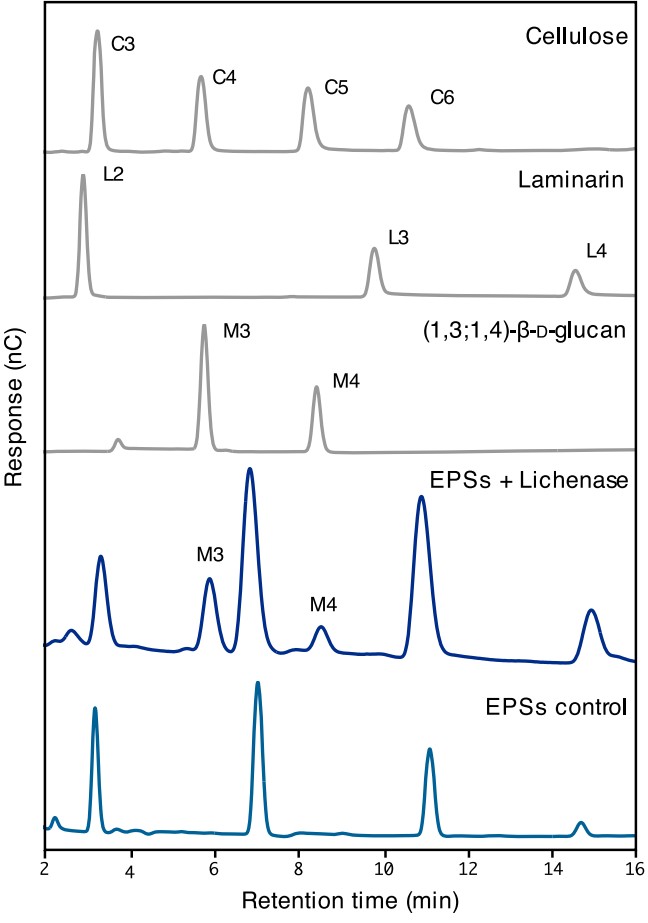

**Fig. 2 | Oligosaccharide profiling of *R. ilealis* CRIB[T] EPS.** *R. ilealis* CRIB[T] EPS was precipitated with ethanol and treated with the (1,3;1,4)-β-D-glucan-specific enzyme lichenase for 24 h at 50 °C. The HPAEC-PAD chromatogram of lichenase-treated EPS (EPS + lichenase) was compared with the control chromatogram without lichenase treatment (EPS control) and with chromatograms of pure standard oligosaccharides. The latter were cellotriose (C3), cellotetraose (C4), cellopentaose (C5) and cellohexaose (C6) derived from cellulose; laminaribiose (L2), laminaritriose (L3) and laminaritetraose (L4) derived from laminarin; and M3 (DP3) and M4 (DP4) derived from (1,3;1,4)-β-D-glucan. Numbers refer to degrees of polymerization (DP). The two peaks from lichenase-treated EPS (EPS + lichenase) have the same retention times as M3 and M4. EPS, exopolysaccharides.

## *R. ilealis* CRIB[T] possesses a gene possibly encoding (1,3;1,4)-β-D-glucan synthase

After confirming (1,3;1,4)-β-D-glucans do exist in *R. ilealis* EPSs, we searched for a possible bacterial (1,3;1,4)-β-D-glucan synthase in the Carbohydrate Active Enzymes database (CAZY, http://www.cazy.org/)[30] and National Center for Biotechnology Information protein database (NCBI) (Supplementary Table 1)[16]. Out of 5 putative GT2 proteins in the *R. ilealis* CRIB[T] database, only one GT2 protein (CED93608.1) contains the polysaccharide synthase signature sequence [D,D,D,Q(Q/R)XRW] signature. We, therefore, named the gene *RiGT2* (*R. ilealis* glycosyltransferase family 2).

## Expression of the *RiGT2* in *Saccharomyces cerevisiae* resulted in the secretion of (1,3;1,4)-β-D-glucans

To characterize the function of the *RiGT2* gene, the gene was cloned into the yeast-enhanced GFP-fusion vector pDDGFP-2 (modified from p424 GAL1, ATCC® 87329™) (Supplementary Fig. 2)[31,32]. The *RiGT2* gene was placed under the control of the *GAL1* promoter, with a Tobacco Etch Virus protease (TEVp)-eGFP-8His sequence added at the *C*-terminus. Membrane-bound *RiGT2* was expressed in the quadruple knockout yeast strain (LoGSA), which exhibits low endogenous glucosyltransferase background activities due to deficient *FKS1*, *GSY1*, and *GSY2* genes[33], and *RiGT2* gene homologs are absent in the yeast genome.

Following *RiGT2* expression, the LoGSA cells were disrupted with glass beads. The membrane fractions were collected from the total cell lysate, and the membrane proteins were separated using reducing SDS–PAGE. This resolved a ~100 kDa protein band detected by in-gel fluorescence (Supplementary Fig. 3a), which corresponds roughly to the predicted molecular weight of *RiGT2* protein with *C*-terminal GFP (102.7 kDa) (Supplementary Fig. 4). Total *RiGT2* protein expression was determined by whole-cell fluorescence calibrated against a GFP standard[31], and most colonies screened produced ~1 mg of *RiGT2*-GFP per liter culture (Supplementary Table 2). The identity of the *RiGT2* protein was confirmed by LC–MS coupled with in-gel trypsin digestion (Supplementary Figs. 3b and 5).

We hypothesized that *RiGT2* is involved in (1,3;1,4)-β-D-glucan biosynthesis, and hence, LoGSA expressing the *RiGT2* gene is expected to produce (1,3;1,4)-β-D-glucans. The success of this approach relies heavily on the correct folding of the recombinant *RiGT2*, and the accessibility of the UDP-glucose and ancillary proteins, if any, that are needed for (1,3;1,4)-β-D-glucans synthesis in LoGSA. We then determined if synthesized (1,3;1,4)-β-D-glucans were present in the *RiGT2* transgenic LoGSA EPS preparation (alcohol-insoluble residue, AIR) using oligosaccharide profiling after lichenase treatment (Fig. 1). We detected (1,3;1,4)-β-D-glucan with a DP3/DP4 molar ratio of 15:1 and no (1,3;1,4)-β-D-glucan oligosaccharides were detected in the EPS preparation (AIR) of control LoGSA carrying empty vector (EV) (Fig. 3, Supplementary Fig. 6). Compared to the oligosaccharide profile of the EPSs isolated from *R. ilealis* CRIB[T], the (1,3;1,4)-β-D-glucan synthesized by *RiGT2* LoGSA had a much higher DP3:DP4 ratio (15:1) than the (1,3;1,4)-β-D-glucan in *R. ilealis* EPS (2:1). MS analysis of the purified oligosaccharides from *RiGT2* LoGSA gave ions with *m/z* [M+Na]⁺ 527, corresponded to DP3 (Hex₃) (Fig. 4a, b). The result was further confirmed by linkage analysis of the DP3 oligosaccharides (Fig. 4c, Supplementary Fig. 7). The oligosaccharide alditol resulting from the NaBD₄ reduction of DP3 was found to have a 1:1:1 ratio of 3-linked glucitol (reduced reducing end): 4-linked Glc*p*: t-Glc*p* (non-reducing end) (Supplementary Table 3), providing strong evidence for a DP3 structure of β-Glc*p*-(1→4)-β-Glc*p*-(1→3)-β-Glc (G4G3G). Moreover, a specific monoclonal antibody against (1,3;1,4)-β-D-glucan (BS400-3), was used with indirect immunofluorescence microscopy to locate (1,3;1,4)-β-D-glucan (Fig. 5). The confocal microscopy images showed that the monoclonal antibody targeting (1,3;1,4)-β-D-glucan

a

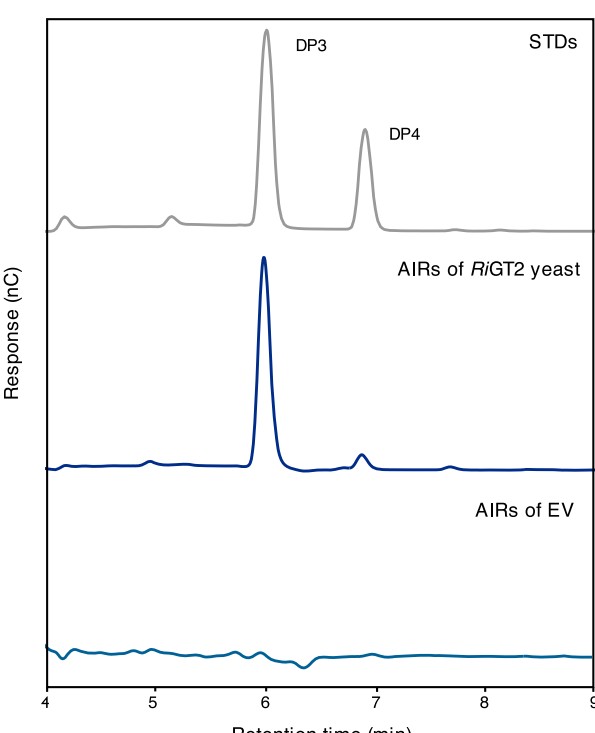

b

**Fig. 3 | Oligosaccharide profiling of *RiGT2* LoGSA by HPAEC-PAD and MALDI-TOF MS. a** HPAEC-PAD chromatograms of the pure oligosaccharide standards (STDs) DP3 and DP4 compared with chromatograms of the lichenase hydrolysates of *RiGT2* LoGSA and the empty vector (EV) LoGSA. The lichenase hydrolysate of AIR prepared from *RiGT2* LoGSA has an abundant peak with a retention time corresponding to the DP3 standard oligosaccharide and a minor peak corresponding to the DP4 standard oligosaccharide, giving a DP3/DP4 ratio of ~15:1. **b** MALDI-TOF spectrum of the *RiGT2* LoGSA lichenase hydrolysate. The signal at *m/z* 527 corresponds to DP3, whereas the signal from DP4 is undetectable. EV empty vector, AIRs alcohol insoluble residues, STDs oligosaccharide standards, DP degrees of polymerization.

labeled only the induced *RiGT2* LoGSA cells (Fig. 5c), but was absent in non-induced LoGSA cells (Fig. 5i). The same conclusion was reached using the same antibody with indirect immunogold microscopy using a colloidal-gold-labeled secondary antibody in conjunction with transmission electron microscopy (TEM). Intense colloidal-gold labeling was found only over the cytoplasm of induced *RiGT2* LoGSA cells (Supplementary Fig. 8).

To further verify the specific activity of *RiGT2*, the microsomal fractions (MFs) from *RiGT2* LoGSA were isolated, and glucan synthase activity was measured by the radiometric in vitro assay[34], using UDP-[$^{14}$C]Glc as a radiotracer. The reactions were stopped by using an ethanol solution (66% aqueous ethanol) and the alcohol-insoluble polymers collected by filtration and the incorporated [$^{14}$C]Glc quantified by scintillation counting. Figure 6a shows that alcohol-insoluble glucan polymers were synthesized in *RiGT2*-containing MFs in the presence of UDP-[$^{14}$C]Glc and the [$^{14}$C]-labeled polymers were degradable by lichenase, suggesting (1,3;1,4)-β-ᴅ-glucans were synthesized in *RiGT2*-containing MFs in vitro. Our gain-of-function yeast model successfully expressed the *RiGT2* gene and was shown to encode the biosynthetic enzyme essential for (1,3;1,4)-β-ᴅ-glucan biosynthesis.

## Structure analysis of theoretical *RiGT2* models

To investigate the possible function of *RiGT2* based on structural homology with other characterized GT2 enzymes, we generated two theoretical *RiGT2* models using both non-template-based AlphaFold2[35] modeling and template-based modeling with SWISS-MODEL[36]. The theoretical model generated by AlphaFold2, *RiGT2*$_A$, showed a high degree of reliability, as indicated by an overall high per-residue

confidence metric (pLDDT; predicted local distance difference test) of 92.1, and a pTM score (predicted template modeling score) of 0.895. *RiGT2* contains 637 amino acids that are predicted to have high fold similarity to *C. sphaeroides* cellulose synthase catalytic subunit *Cs*BcsA (PDB 4HG6) (Fig. 7a, b). Two regions in the theoretical *RiGT2* model displayed low or very low confidence scores (pLDDT < 70), namely residues 434–450 and the *C*-terminal region 609–637. Due to the poor prediction quality of the *C*-terminal region, residues 612–637 were deleted from the model. In *Cs*BcsA, this region forms a long extension originating from the PilZ domain that does not interact directly with the polysaccharide-binding channel. Calculated on the structure-adjusted sequence alignment of the *Cs*BcsA and *RiGT2* sequences, the pair-wise sequence identity is 25.7%.

Although AlphaFold2 is a highly reliable tool for generating accurate structural models, we anticipated that the model generated by AlphaFold2 would be biased towards bacterial cellulose synthase *Cs*BcsA (PDB 4HG6) due to the larger amount of experimental data available for cellulose synthases. Therefore, we generated a second model, *RiGT2*$_S$, using SWISS-MODEL based on a single template, plant (1,3;1,4)-β-ᴅ-glucan synthase *Hv*CslF6 (PDB 8DQK[37]), to provide complementary information to the *RiGT2*$_A$ model. The two models have inherited features from their respective parental structures. In the case of the AlphaFold2 model, this is not from a single template structure but the result of AlphaFold2 training on all available homologous templates available in the Protein Data Bank, which currently is dominated by bacterial cellulose synthase homologs. The *RiGT2* model generated by SWISS-MODEL using *Hv*CslF6 as a template has severe template bias, which is expected when only one template is used. Therefore, any similarities or differences observed for the two

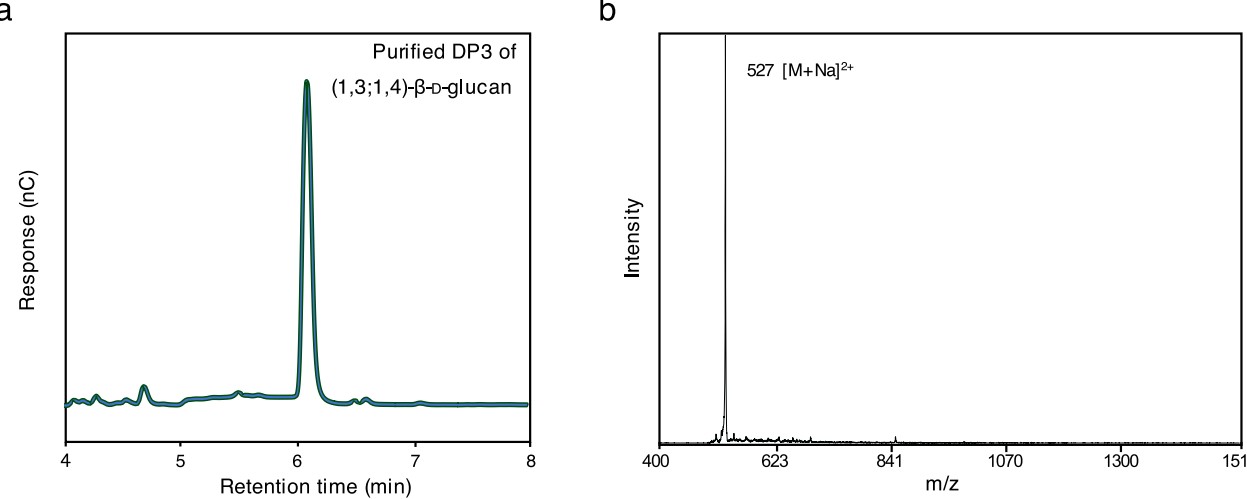

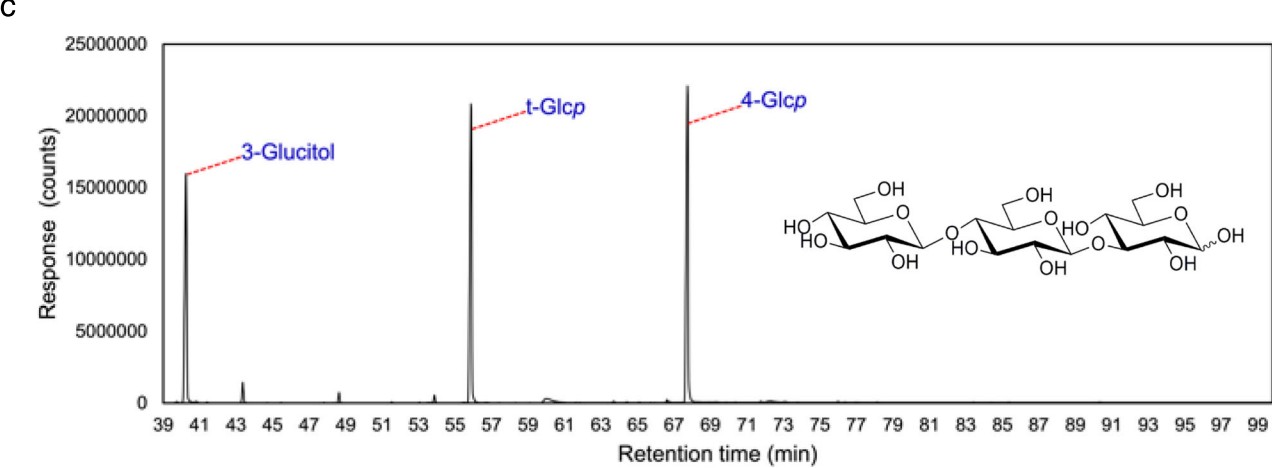

**Fig. 4 | Linkage analysis of the DP3 (1,3;1,4)-β-ᴅ-glucan oligosaccharide from *RiGT2* LoGSA. a** HPAEC-PAD chromatogram of the purified DP3 oligosaccharide derived from a lichenase digest of *RiGT2* LoGSA. **b** MALDI-TOF spectrum shows a single ion with an *m/z* 527 that corresponds to the molecular weight of DP3 (G4G3G). **c** The oligosaccharide was reduced by NaBD₄, followed by GC–MS analysis of the partially methylated alditol acetate (PMAA) derivatives prepared from the sample. The total ion current (TIC) chromatogram shows the presence of 3-linked glucitol (3-glucitol) from the reduced reducing end, terminal glucopyranose (t-Glc*p*) from the non-reducing end, and 4-linked glucopyranose (4-Glc*p*) as the residue in the middle of the trisaccharide chain.

theoretical models (Supplementary Fig. 9) will in most parts only reflect the similarities and differences between the experimental structures of *Cs*BcsA and *Hv*CslF6.

The *Ri*GT2ₐ model displays most of the features of *Cs*BcsA with an extramembrane catalytic glycosyltransferase (GT) domain, a transmembrane (TM) domain, and a *C*-terminal PilZ domain (Fig. 7a, b and Supplementary Fig. 10). The GT domain features seven β-strands and six α-helices and contains the [D,D,D,Q(Q/R)XRW] signature found in GT2 polysaccharide synthases[38] (Fig. 7c and Supplementary Fig. 10). The first aspartate residue of the motif is Asp139, which has been assigned a role in coordinating the uridine ring in the UDP-Glc donor. The next aspartate of the motif is Asp188 in the DXD motif ([186]DA**D**[188]). Asp188 is expected to stabilize the diphosphate group of the donor via the metal cofactor (Mg²⁺), whereas Asp186 is predicted to stack with the ribose C4′–C5′ moiety of the donor's UDP moiety. The last aspartate Asp280 is positioned in the conserved TED motif and is predicted to act as a catalytic base. The [Q(Q/R)XRW] motif is defined by [316]QRDRW[320] in *Ri*GT2. Based on the *Cs*BcsA structure, Trp320 in *Ri*GT2 could stack with a polysaccharide chain as Trp383 stacks with Glc-17 at the acceptor site in *Cs*BcsA (PDB 4HG6[19]; PDB 4P00 and 4P02[20]; PDB 5EJ1 and 5EJZ[18]).

The TM domain in *Ri*GT2ₐ is predicted to contain seven TM helices (TMH2–8 in *Cs*BcsA) that form a polysaccharide-binding channel. The model also includes three interface helices (IFH1–3). *Ri*GT2 lacks the *N*-terminal TMH that coils with the periplasmic *Cs*BcsB subunit in the *Cs*BcsA–*Cs*BcsB complex[19]. In the *C. sphaeroides* genome (GenBank CP000143.2), the genes coding for the catalytic subunit BcsA (gene RSP_0333) and the BcsB subunit (gene RSP_0332) are located in an operon, whereas in the *R. ilealis* genome (GenBank LN555523.1), the gene coding for *Ri*GT2 (gene CRIB_856) does not cluster with a corresponding *bcsB* gene. Thus, based on the amino-acid sequence and the gene organization, *Ri*GT2 is not likely to depend on a BcsB subunit, at least not of the same type as *C. sphaeroides* and related bacterial sugar polymerases where BcsB is required for polysaccharide synthesis[39,40]. In this respect, *Ri*GT2 is more like the homotrimeric plant CesA7[41] and CesA8[42] which do not require a BcsB subunit, and whose catalytic domains also lack the *N*-terminal TMH. However, the trimeric plant enzymes instead have an extra TMH at the *C*-terminus that participates in oligomerization into homotrimers.

Bacterial sugar polymerases that produce biofilm polysaccharides contain a *C*-terminal PilZ domain that serves as an effector subunit that binds the secondary messenger cyclic-di-GMP for intracellular

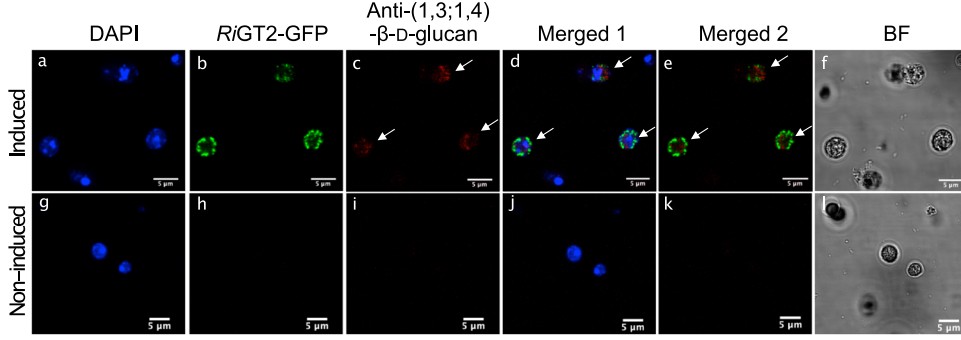

**Fig. 5 | Confocal microscopy of *Ri*GT2-GFP and indirect immunofluorescence microscopy of (1,3;1,4)-β-D-glucans.** Images of induced (**a**–**f**) and non-induced (**g**–**l**) *Ri*GT2 LoGSA were obtained using an inverted confocal microscope. The *Ri*GT2 protein expressed in *Ri*GT2 transgenic yeast containing *C*-terminal GFP fluoresced green, but no fluorescence was shown in non-induced cells (*Ri*GT2-GFP). The location of the (1,3;1,4)-β-D-glucans was determined by indirect immuno-fluorescence using a mouse monoclonal antibody against (1,3;1,4)-β-D-glucan (BS400-3) and goat anti-mouse IgG (H + L) conjugated to Alexa Fluor™ 555 showing a red fluorescence, which overlapped GFP fluorescence (Merge2), indicating (1,3;1,4)-β-D-glucans were produced only in yeast with *Ri*GT2-GFP expression.

Brightfield and merged fields are also shown for comparison. Representative of three independent experiments. **a**, **g** DNA staining with 1 μg/mL DAPI (blue). **b**, **h** *Ri*GT2-GFP fusion protein (green). **c**, **i** Indirect immunofluorescence detection of (1,3;1,4)-β-D-glucan. **Merged 1**. Merged images of DAPI, *Ri*GT2-GFP, and Anti-(1,3;1,4)-β-D-glucan. **Merged 2**. Merged images of *Ri*GT2-GFP and Anti-(1,3;1,4)-β-D-glucan. **BF** brightfield images of yeast cells. All images have been exposed to the same laser intensity when visualizing the fluorescence. **BF** brightfield. **a**–**f** Induced *Ri*GT2-GFP transformed yeast. **g**–**l** Non-induced *Ri*GT2-GFP transformed yeast. White arrows in panels Anti-(1,3;1,4)-β-D-glucan and Merged 1 and Merged 2 indicate red fluorescence of Alexa Fluor 555.

signaling, and in *C. sphaeroides* cellulose synthase, binding of cyclic-di-GMP stimulates the synthesis of bacterial cellulose[20]. In *Ri*GT2, residues 522–594 are predicted to form a PilZ domain shaped as a six-stranded antiparallel β-barrel similar to that in *Cs*BcsA (residues 584–676). The PilZ region displays very low overall sequence similarity to *Cs*BcsA PilZ (Supplementary Fig. 10), but the PilZ [RXXXR] signature is present in *Ri*GT2 (Supplementary Fig. 11), [517]RTSER[521], as well as the [D/NXSXXG] motif[43], [554]NLSEKG[550].

A conserved gating loop in the GT domain of *Cs*BcsA (residues 499–512) controls access to the active site[20] (Fig. 7a, b and Supplementary Fig. 10). The gating loop in *Cs*BcsA adopts different conformations (open or resting state) in response to the presence or absence of cyclic-di-GMP, where the resting state conformation blocks access to the active site, secured by a salt link between Arg580 in the PilZ domain and Glu371 in the GT domain. In the open state, Arg580 swings 180° to interact with cyclic-di-GMP, thereby breaking the salt link to Glu371. The corresponding gating loop in *Ri*GT2 comprises residues 436–448. In both *Cs*BcsA and *Ri*GT2 the *N*-terminal region of the gating loop includes the signature [FXVTXK] ([439]FNVTLK[444] in *Ri*GT2 and [503]FAVTAK[508] in *Cs*BcsA) while the rest of the residues of the gating loop are non-conserved. In the AlphaFold2 model of *Ri*GT2, the gating loop region is one of two regions with a low confidence score. The residues forming the regulatory salt link in *Cs*BcsA (Arg580 and Glu371) are not fully conserved in *Ri*GT2 and correspond to Arg517 and Thr308, respectively.

The channel in *Cs*BcsA spans -50 Å and accommodates 11 β-1,4-linked D-glucosyl units (glucosyl 8–18 in PDB 4HG6) at the acceptor side. A superposition of the *Cs*BcsA structure with the *Ri*GT2$_A$ and *Ri*GT2$_S$ models shows that *Ri*GT2 would be able to accommodate 11 glucosyl units (glucosyl 8–18 in PDB 4HG6). The key side chains that constitute the individual glucosyl-binding sites were analyzed in *Cs*BcsA, *Ri*GT2, and *Hv*CslF6 (Supplementary Table 4). While some side chains exist in all three proteins, there is no single glucosyl-binding site that is fully conserved. Since *Ri*GT2 and *Hv*CslF6 are both predicted to synthesize (1,3;1,4)-β-D-glucans, we expected the residues of the binding sites in *Ri*GT2 to be more similar to *Hv*CslF6 than to *Cs*BcsA, but *Ri*GT2 has more side chains in common with *Cs*BcsA (Supplementary Table 4).

Since a polysaccharide chain, cellulose or (1,3;1,4)-β-D-glucan, needs to be translocated through the channel during synthesis, the side-chain interactions must not be too specific, and different types of

side chains could fulfill the purpose. The more important structural feature of a channel that accommodates a nascent (1,3;1,4)-β-D-glucan, as opposed to cellulose, is the ability to adjust to the unique structural features introduced by the β-1,3 linkages. Due to the expected bias from cellulose synthase templates, such features are more difficult to analyze in the *Ri*GT2$_A$ model. Indeed, the shape and size of the channel in the *Ri*GT2$_A$ model are suspiciously similar to *Cs*BcsA and appear too narrow to be able to accommodate a glucan with β-1,3 glycosidic bonds at every third linkage. Since the polysaccharide-binding channel is formed mainly by the TMHs, it is likely that the principal means by which to adapt the channel to (1,3;1,4)-β-D-glucan is by repositioning one or several TMHs. Comparison of the TMHs in *Cs*BcsA and *Hv*CslF6 indeed shows that the TMHs are somewhat displaced in *Hv*CslF6 to expand the channel (also inherited by the *Ri*GT2$_S$ model), which could be sufficient to accommodate a (1,3;1,4)-β-D-glucan chain (Supplementary Fig. 12). The slightly more spacious channel in *Hv*CslF6, at least for some subsite, can be also seen by comparing the representations of the molecular surfaces for *Cs*BcsA, *Hv*CslF6 and the theoretical *Ri*GT2 models (Supplementary Fig. 13).

By analyzing the sequences and models, experimental and predicted, for bacterial *Cs*BcsA, *Ri*GT2, and plant *Hv*CslF6, our conclusion is that *Ri*GT2 shares a higher degree of similarity with *Cs*BcsA, but that the precise orientations of the TMHs may be more similar to the TM domain in *Hv*CslF6. Thus, while the theoretical models cannot offer proof of *Ri*GT2 synthesizing (1,3;1,4)-β-D-glucan, we find nothing that argues against this function at the structural level. However, the predicted structural features of *Ri*GT2 would also be compatible with the synthesis of cellulose.

### Evaluation of the PilZ domain in *Ri*GT2 (1,3;1,4)-β-D-glucan biosynthesis

The similarities between *Cs*BcsA and *Ri*GT2 at the sequence level strongly suggest *Ri*GT2 is also regulated by a PilZ domain. To the best of our knowledge, c-di-GMP is absent from the yeast secondary messenger system, but the *Ri*GT2 LoGSA cells could still synthesize (1,3;1,4)-β-D-glucan. This apparent discrepancy required further investigation and the effect of different c-di-GMP concentrations on *Ri*GT2 activity was examined by incubating *Ri*GT2 LoGSA microsomal preparations with c-di-GMP (30, 60, or 90 μM) and UDP-[¹⁴C]Glc, and monitoring the (1,3;1,4)-β-D-glucan synthase activity with a radiometric assay. Surprisingly, there was no increase in synthase activity with an

a

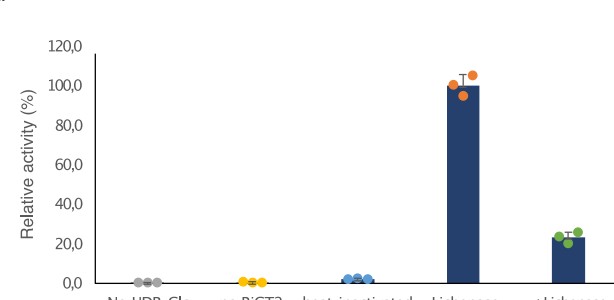

b

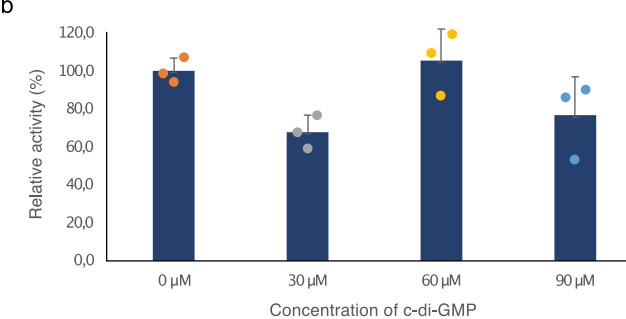

**Fig. 6 | The specific activity of the *Ri*GT2-containing microsomal fraction by radiometry assay.** Microsomal fractions (MF) from both *Ri*GT2 and empty vector (EV) LoGSA were incubated with MgCl₂, the substrate UDP-Glc and ¹⁴C-labeled UDP-Glc as a radiotracer. The reactions were terminated by 66% EtOH and the alcohol-insoluble fractions were filtered and dried for scintillation counting. **a** The reactions were performed without UDP-Glucose (no UDP-Glc), without MFs (no *Ri*GT2), with MFs heated to 95 °C for 10 min (heat inactivated), without lichenase (−Lichenase) and with lichenase (+Lichenase). The activity of *Ri*GT2-containing fractions was normalized by the activity from EV MF and relative to the reaction in the absence of lichenase (−lichenase). **b** Additional c-di-GMP at different concentrations (30, 60, and 90 μM) was added to the reaction mixtures. Data are presented as mean values ± SD. Each data point represents an independent experiment. The error bars indicate the standard deviation (SD) calculated from three replicates (*n* = 3).

increase in c-di-GMP concentration (Fig. 6b), showing that (1,3;1,4)-β-ᴅ-glucan synthesis is not dependent on the presence of c-di-GMP. To further verify the direct binding of c-di-GMP to the PilZ domain of *Ri*GT2, we expressed and purified the recombinant PilZ domain of *Ri*GT2 with *C*-terminal GFP fusion protein (*Ri*PilZ-GFP) and performed in vitro binding studies using Isothermal Titration Calorimetry (ITC) (Supplementary Fig. 14a) and Surface Plasmon Resonance (SPR) (Supplementary Fig. 14b), but no direct binding was detected. This lack of binding could be due to the requirement of interactions provided by the absent GT domain or interference from possible glycosylation in the yeast system. Alternatively, it could indicate that *Ri*GT2 mediates the synthesis of (1,3;1,4)-β-ᴅ-glucan through a different mechanism that does not strictly require the binding of c-di-GMP.

To further verify that the PilZ domain is necessary for *Ri*GT2 activity, we generated a *Ri*GT2 with the PilZ domain deleted (*Ri*GT2-*PilZD*) and transformed the gene into LoGSA. Interestingly, after digesting with lichenase the AIR prepared from transgenic yeast expressing the *C*-terminal truncated *Ri*GT2, we found no (1,3;1,4)-β-ᴅ-glucan was synthesized (Supplementary Fig. 15a). This result was further supported by using the radiometric in vitro assay (Supplementary Fig. 15c), and we concluded that no (1,3;1,4)-β-ᴅ-glucan synthase activity was detected when the PilZ domain was cropped from *Ri*GT2. Despite no interaction between the *Ri*GT2 PilZ domain and the secondary messenger c-di-GMP, the presence of the *C*-terminal PilZ domain remains crucial for *Ri*GT2 (1,3;1,4)-β-ᴅ-glucan synthase activity, possibly by stabilizing the protein structure or regulating enzymatic activity through conformational changes.

## Phylogenetic analysis of (1,3;1,4)-β-ᴅ-glucan synthases

The phylogenetic tree was generated for (1,3;1,4)-β-ᴅ-glucan synthases and cellulose synthases of cereals and other grasses, and for Gram-positive and Gram-negative bacteria (Supplementary Fig. 16a). Each clade is very distant from each another. In a previous study, it was suggested that putative BgsA (1,3;1,4)-β-ᴅ-glucan synthases from Gram-negative bacteria are more likely to have evolved from curdlan (a 1,3-β-ᴅ-glucan) synthase (CrdS)[8], whereas the grass (1,3;1,4)-β-ᴅ-glucan synthases CslF6, CslH, and CslJ are phylogenetically closer to cellulose synthase CesA[44]. However, there have been only a few phylogenetic studies on the GT2s of Gram-positive bacteria. Scott et al. (2020) reported a series of putative *Clostridial cellulose synthase* (*Ccs*) operons among the class Clostridia[45]. Information from this study could allow us to identify possible conserved regions that may be critical for bacterial (1,3;1,4)-β-ᴅ-glucan synthases, being evolved to mediate the synthesis of the two linkages. To this end, homologous sequences were obtained using a BLAST server with *Clostridioides difficile* CcsA (NCBI accession: WP_021391996.1) and *Ri*GT2 (CED93608.1 or WP_180703307.1) sequences as individual queries. The identity of all selected sequences was greater than 30%, with over 70% query coverage. A neighbor-joining likelihood phylogenetic tree was then generated for a non-redundant subset of 188 sequences from *Ri*GT2 and CcsA homologous (Supplementary Fig. 16b). The sequences are clustered into three main clades: clade 1 includes *Ri*GT2 (1,3;1,4)-β-ᴅ-glucan synthase; clade 2 remains unknown, but with three proteins annotated as cellulose synthases (POB11864.1, RSU08770.1, GAF20063.1); and clade 3 contains three putative CcsAs[45] (Supplementary Fig. 16b). Clade 1 sequence, which contains *Ri*GT2, were therefore selected as (1,3;1,4)-β-ᴅ-glucan synthase candidates and renamed Cgs (Clostridia β-glucan synthase) because most of the sequences are in species in the class Clostridia (Fig. 8).

To identify conserved motifs that may play critical roles in bacterial (1,3;1,4)-β-ᴅ-glucan synthase, we aligned sequences from five Cgs and four CcsA proteins. The protein alignment reveals three possible conserved sequences in Cgs, marked in the red boxes in Supplementary Fig. 17. We can only speculate that these three marked regions could be bacterial (1,3;1,4)-β-ᴅ-glucan conserved motifs because this is the only bacterial (1,3;1,4)-β-ᴅ-glucan synthase that has been biochemically characterized in a Gram-positive bacterium. A conserved sequence referred to as "switch motif" is present in interface helix 3 (IFH3) of the plant (1,3;1,4)-β-ᴅ-glucan synthase *Hv*CslF6[37]. It was confirmed that Tyr787 in the switch motif of *Hv*CslF6 was critical for introducing β-1,3 glycosidic bonds within (1,3;1,4)-β-ᴅ-glucan[37]. However, this motif is not present in *Ri*GT2 or *Cs*BcsA, where Tyr787 in *Hv*CslF6 corresponds to Leu424 and Leu487, respectively. Therefore, further research is needed to determine whether IFH3 in Cgs could also regulate the ratio of 1,3 and 1,4 linkages in bacterial (1,3;1,4)-β-ᴅ-glucans.

## (1,3;1,4)-β-ᴅ-glucans and (1,3;1,4)-β-ᴅ-glucan synthases are present in other Gram-positive bacteria species

The Gram-positive bacteria *Clostridium bornimense*, *Clostridium ventriculi*, *Robinsoniella peoriensis* and *Clostridium tyrobutyricum* were chosen to investigate if they contain (1,3;1,4)-β-ᴅ-glucan and (1,3;1,4)-β-ᴅ-glucan synthases based on the following: (1) they have *Ri*GT2 homologs with high sequence similarities; (2) these species are commercially available, and (3) out of six species that are available, the four selected could be cultured in our anaerobic system, whereas others (*Clostridium nigeriense* and *Niameybacter massiliensis*) either grew very slowly or not at all and we were unable to harvest enough EPSs for the subsequent analyses. All these bacteria have identified *Ri*GT2 homologs with high sequence homology and are positioned in the phylogeny within in same order Eubacteriales and class Clostridia. EPSs were isolated from the four selected species, and treated with lichenase, and the resulting oligosaccharides were analyzed by HPAEC-PAD. (1,3;1,4)-

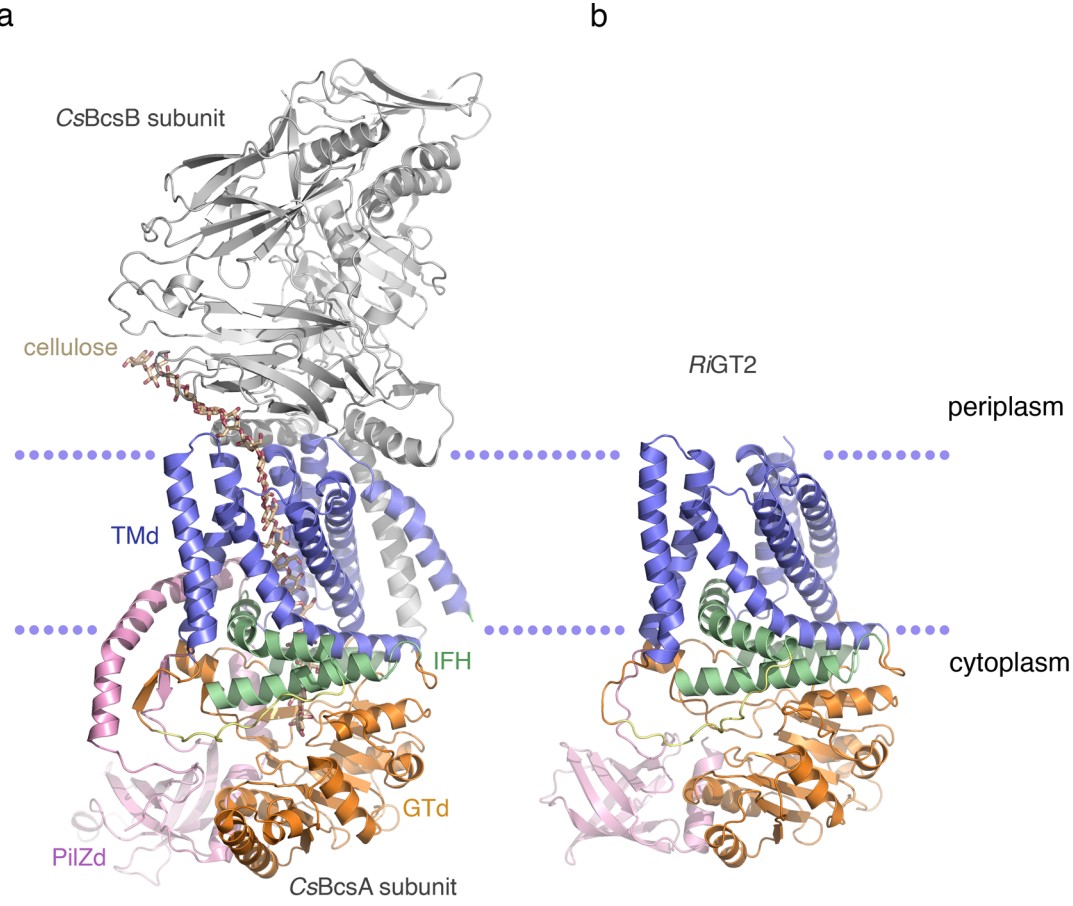

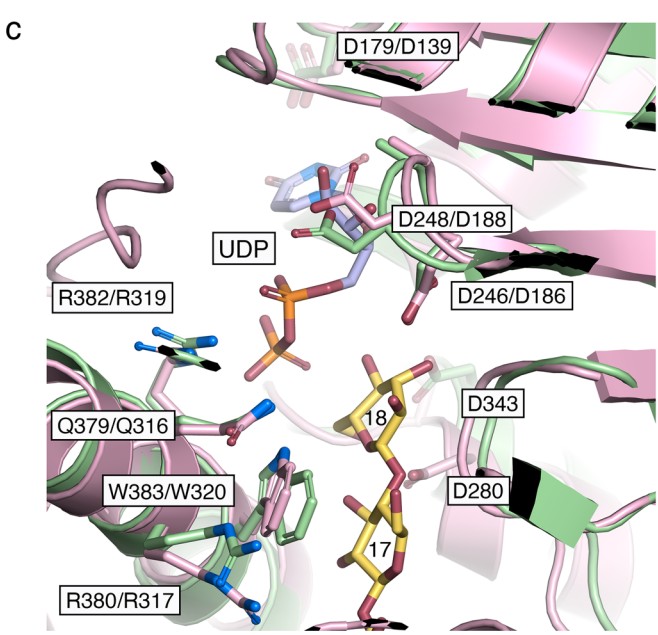

β-D-glucan DP3 and (1,3;1,4)-β-D-glucan DP4 were found in the hydrolysates from *C. ventriculi, R. peoriensis*, and *C. tyrobutyricum* in various ratios, whereas the hydrolysate from *C. bornimense* contained only (1,3;1,4)-β-D-glucan DP3 (Figs. 9 and S18). This result indicates that (1,3;1,4)-β-D-glucans are more widespread as components of EPSs than was previously thought, and the DP3:DP4 ratio may differ between species.

## Discussion

EPSs of Gram-positive bacteria are crucial for their biofilm formation, cell adhesion, and pathogenicity. Many polysaccharides have been identified as EPSs, such as teichoic acid (TA) and poly-β-1,6-linked *N*-acetyl glucosamine (PNAG/PIA), and serve different functions[46]. However, no previous report has shown that (1,3;1,4)-β-D-glucans are part of the EPSs of Gram-positive bacteria. Here, we report (1,3;1,4)-β-D-

**Fig. 7 | Comparison of *Cs*BcsA and the AlphaFold2 model of *Ri*GT2. a** Ribbon representation of the crystal structure of the *C. sphaeroides* cellulose synthase complex consisting of BcsA and BcsB (PDB 4HG6 [https://www.rcsb.org/structure/4HG6]), and **b** the predicted AlphaFold2 model of *Ri*GT2. The domains in the *C. sphaeroides* BcsA-BcsB complex are colored as follows: *Cs*BcsA GT domain (GTd), orange; TM domain (TMd), blue; interface helices (IFH), green; PilZ domain (PilZd), pink; *Cs*BcsB, gray; cellulose chain, yellow; and the gating loop connecting interface helix 3 and the PilZ domain is colored yellow. The cellulose chain runs from the center of the GT domain in BcsA and exits the channel at the BcsA-BcsB interface at the periplasmic face. The membrane bilayer has been delineated by a blue dotted line. The same colors were used for *Ri*GT2. **c** Overlay of the active site in *Cs*BcsA (green; PDB 4HG6) and the *Ri*GT2$_A$ model (pink). The view is rotated 180° with respect to panel (**b**). The two terminal glucosyl units Glc-17 and Glc-18 (yellow) and the UDP molecule (light blue with orange phosphate groups) in the *Cs*BcsA/BcsB complex are shown. The residues that are part of the [D,D,D,Q(Q/R)XRW] signatures are shown, see text for details. Residue names are shown as pairs where the first residue corresponds to that in *Cs*BcsA and the second to the equivalent residue in *Ri*GT2. (Schrödinger, L. & DeLano, W., 2020. PyMOL, Available at: http://www.pymol.org/pymol).

glucans as a novel component of the EPS of *R. ilealis* CRIB$^T$. When analyzed in secreted EPS, the (1,3;1,4)-β-ᴅ-glucans contained primarily DP3 and DP4 subunits, and their structures resemble those of grass (1,3;1,4)-β-ᴅ-glucans[1,7,47,48]. Rather surprising however, (1,3;1,4)-β-ᴅ-glucans produced by *RiGT2* transgenic yeast consisted mainly of the DP3 (G4G3G) subunit. Our study suggests that the (1,3;1,4)-β-ᴅ-glucan produced by *Ri*GT2 is structurally more closely related to lichenin (DP3 dominant) than to the (1,3;1,4)-β-ᴅ-glucans of Gram-negative bacteria *S. meliloti* (DP2 only). The discrepancy of DP3:DP4 ratio could be that the recombinant *Ri*GT2 in the LoGSA cells are not in their native lipid environment, or the *C*-terminal fusion GFP could also impact the conformation and function of *Ri*GT2. Lee and Hollingsworth (1997) reported that another Gram-positive bacterium, *C. ventriculi*, contains β-glucans with unique linkages[49]. We have further confirmed by phylogenetic analysis and polysaccharide analysis that *C. ventriculi* produce (1,3;1,4)-β-ᴅ-glucans and contains potential (1,3;1,4)-β-ᴅ-glucan synthases. Indeed, phylogenetic analysis showed the widespread occurrence of *Ri*GT2 homologs in Gram-positive bacteria, especially in the Clostridiaceae family. However, we cannot rule out that other non-homologous enzymes are also involved in (1,3;1,4)-β-ᴅ-glucan biosynthesis. This has been shown in the biosynthesis of poly-β-1,6-*N*-acetylglucosamine (PNAG). For example, bacterial PNAG in *Staphylococcus epidermidis* is synthesized by the *icaADBC* operon[50] whereas the PNAG in *Bacillus subtilis* is synthesized by the non-homologous *epsHIJK* operon[51,52].

Cyclic-di-GMP is a potent inducer for biofilm formation in bacteria[51,52]. It activates the cellulose (BcsA-B) synthase complex and the poly-β-1,6-linked *N*-acetyl glucosamine synthase (Pga) for biofilm production[39]. In the absence of c-di-GMP, the resting state of the gating loop (residues 499–517) is stabilized by a conserved salt link in *Cs*BcsA (PilZ Arg580-GT Glu371)[19]. The c-di-GMP-bound state of *Cs*BcsA has a c-di-GMP dimer bound in the PilZ domain with the gating loop in an open conformation that allows entry to the substrate-binding pocket (PDB 4P00 and 4P02[20]). The transition from the resting to open state of the gating loop is controlled by conformational changes that break the salt link, more specifically, the two arginine residues of the RXXXR motif in the PilZ domain (Arg580 and Arg584) rearrange to stack with two of the guanylate rings in c-di-GMP monomer A, and at the same time form hydrogen bonds to the N7 and O6 atoms of the two guanylate rings of in c-di-GMP monomer B. The corresponding arginine residues in *Ri*GT2 (Arg517 and Arg521) are conserved and are expected to be able to interact with a c-di-GMP dimer in a similar way in an open state (Supplementary Fig. 11).

Based on the theoretical *Ri*GT2$_A$ model, Thr308 occupies the position of Glu371 in *Cs*BcsA. Thr308 cannot form a salt link with Arg517, but a nearby glutamate residue, Glu520, could possibly provide this opportunity in a resting state of *Ri*GT2. Interestingly, heterologously expressed *Ri*GT2 can still synthesize (1,3;1,4)-β-ᴅ-glucan in the absence of c-di-GMP in the yeast system. The activity of *Ri*GT2 was independent of c-di-GMP concentration based on a radiometry assay. Additionally, no binding of c-di-GMP to the purified *C*-terminal PilZ domain of *Ri*GT2 was detected in our SPR and ITC binding studies. These results indicate that *Ri*GT2-mediated (1,3;1,4)-β-ᴅ-glucan synthesis may be independent of c-di-GMP activation. However, since the *Ri*GT2 PilZ domain has the key residues for c-di-GMP binding, we can not rule out the possibility of in vivo transcriptional regulation by c-di-GMP as, for example, the *psl* genes encoding Psl polysaccharide synthesizing machinery in *P. aeruginosa* are positively regulated by c-di-GMP at the transcriptional level[53,54].

The functions of (1,3;1,4)-β-ᴅ-glucans can vary from species to species. Due to the distribution of β-1,3 linkages, "kinks" in the structure make (1,3;1,4)-β-ᴅ-glucans unable to lay precisely on each other and are thus more flexible and soluble compared to other linear polysaccharides[55]. (1,3;1,4)-β-ᴅ-glucan found in the cell walls of cereals and other grasses plays an essential role in cell expansion and in the mobilization of endosperm cell walls[56]. In fungi and charophycean green and brown algae, (1,3;1,4)-β-ᴅ-glucan has a significant architectural role in supporting cell wall structure[5,57]. It has also been proposed that the bacterial (1,3;1,4)-β-ᴅ-glucan secreted by *S. meliloti* is associated with the bacterium's attachment to plant roots, which is consistent with bacterial biofilm formation mediating cell-to-cell and cell-to-surface interactions[8]. This, therefore, indicates that bacterial (1,3;1,4)-β-ᴅ-glucan may contribute to bacterial autoaggregation and biofilm production.

Some Gram-positive bacteria have been identified as pathogens with increased antimicrobial resistance because of their biofilm formation. The matrix of many known biofilms is composed predominately of cellulose, with PNAG and alginate as less abundant components[58]. The discovery of (1,3;1,4)-β-ᴅ-glucans in bacterial EPS introduces an important intervention target. More importantly, knowledge of the genes and proteins involved in EPS biosynthesis is fundamental to the study of biofilm formation and control[58]. The presence of (1,3;1,4)-β-ᴅ-glucans in the EPS of groups of bacterial species has been overlooked in the past, and knowledge is limited about their involvement in biofilm formation.

In summary, we have identified a (1,3;1,4)-β-ᴅ-glucan synthase in the Gram-positive bacterium *R. ilealis* CRIB$^T$. Using the yeast platform (Fig. 1), we have provided unequivocal direct gain-of-function evidence for the participation of the prokaryotic *RiGT2* gene in (1,3;1,4)-β-ᴅ-glucan biosynthesis. For a long time, it was believed that (1,3;1,4)-β-ᴅ-glucans did not exist in bacteria and were present only in plants and fungi (including lichens). It was not until (1,3;1,4)-β-ᴅ-glucans with repetitive DP2 units were found in bacterial species[8] that it was realized that (1,3;1,4)-β-ᴅ-glucans are likely to participate in biofilm formation due to their gel-like physiochemical properties. Hence, identifying crucial biofilm biosynthetic enzymes can provide enzymatic inhibitory targets for the control of biofilm during infections. In addition, manipulation of the (1,3;1,4)-β-ᴅ-glucan synthase could be tuned to produce glycopolymers of desired quality, adding to the burgeoning field of biomaterial engineering towards more targeted physical and biological properties than their natural counterparts[59,60].

To date, yeast has been one of the most successful heterologous overexpression systems in producing membrane proteins for high-resolution structural studies[31]. Having successfully expressed functional *Ri*GT2, the overarching goal of our laboratory is to resolve the 3D structure of this polysaccharide synthase in order to understand its (1,3;1,4)-β-ᴅ-glucan biosynthesis at atomic resolution.

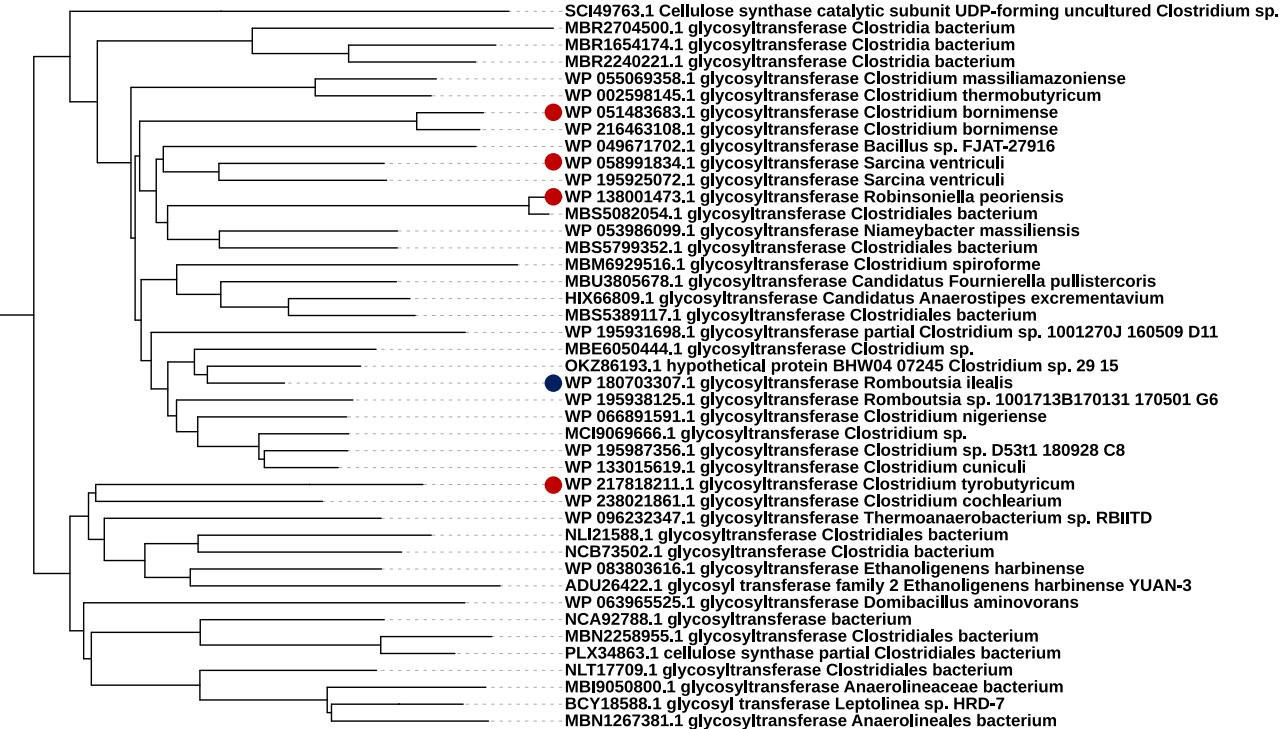

Tree scale: 0.1

**Fig. 8 | The phylogenetic analysis showing the possible homologs in other species of Gram-positive bacteria.** Neighbor joining-likelihood phylogenetic analysis of (1,3;1,4)-β-D-glucan synthase candidates by MEGA version X[63, 64]. The midpoint-rooted phylogenetic tree shows two sub-clades. Sequence entries of all candidates are listed in Supplementary Table 4. *R. ilealis* CRIB[T] (1,3;1,4)-β-D-glucan synthase is marked with a blue dot. Species selected for subsequent (1,3;1,4)-β-D-glucan analysis are marked with red dots. Bar scale indicates distance.

## Methods

### Bioinformatic analysis
All the protein sequences were searched by BLASTP in the National Center for Biotechnology Information database (NCBI) with *Clostridioides difficile* CcsA (WP_021391996.1) and *Ri*GT2 (CED93608.1) sequences as individual queries for 500 target sequences. CD-HIT was used to decrease the redundancy[61]. The protein alignment and phylogenetic analysis were conducted using the EMBL-EBI web tool[62] and MEGA version X[63,64]. The trees were constructed using the neighbor-joining likelihood method based on the Jones–Taylor–Thornton (JTT) model with the parameters as follows: a test of phylogeny, bootstrap method with 1000 replications; substitution type, amino acid; rates among sites, uniform rates. The trees were edited using iTOL online tool[65]. The alignment in Supplementary Fig. 10 was visualized using ESPript3.075 (https://espript.ibcp.fr/ESPript/ESPript/)[66].

### Anaerobic cultivation of *R. ilealis* CRIB[T]
The bacterium *R. ilealis* CRIB[T] (DSM 25109) was purchased from DSMZ GmbH (Leibniz Institute DSMZ-German Collection of Microorganisms and Cell Cultures GmbH, Germany). The inoculation procedures followed the instructions from DSMZ with some modifications. First, 50 mL PYG medium (10 g/L peptone, 5 g/L yeast extract, 5 g/L glucose, 0.5 g/L L-cysteine hydrochloride, 0.2 g/L $Na_2CO_3$, 1.5 g/L $KH_2PO_4$, 0.1 g/L $MgSO_4$, 0.25 mg/L hemin, 0.5 mL/L resazurin, pH 7) was transferred to 125 mL Wheaton® serum bottles (DWK Life Sciences, Millville, NJ, USA) and covered with blue butyl rubber stoppers and crimp caps to prevent gas leakage. After autoclaving (20 min at 121 °C), the atmosphere in the bottles was exchanged by 20 cycles of alternating vacuum and purging (45 s per each cycle) with ultrapure $N_2$ gas (N5.0) through a sterile syringe filter (0.2 μm polyethersulfone (PES)) to ensure an anaerobic environment. Next, the vial of freeze-dried bacteria from DSMZ was opened in a vinyl anaerobic chamber (Coy Laboratory Products, TG Instruments, Helsingborg, Sweden) containing 5% $H_2$, 10% $CO_2$, and 85% $N_2$. The cell pellet was resuspended in medium (10 mL) and 0.1 mL was used to aseptically inoculate the prepared serum bottle that was then taken out of the anaerobic chamber, purged with 80% $N_2$ and 20% $CO_2$ (5 cycles) to provide $CO_2$ for growth, and incubated at 37 °C for 3–5 days on an orbital shaker (100 rpm).

### Stable transformation of *RiGT2* and *RiGT2-PilD* in the LoGSA yeast strain
The PCR primers (Supplementary Table 5) were used to amplify fragments containing full-length open-reading frames of *RiGT2* from synthetic cDNA prepared in pMA plasmid (GeneArt, ThermoFisher Scientific, USA). The cDNA of *RiGT2* or the PilZ domain mutant (*RiGT2-PilD*) were cloned in the pDDGFP-2 vector by homologous recombination, transformed into the quadruple knockout yeast *Saccharomyces cerevisiae* LoGSA[33] and expressed following the method described in Newstead et al.[32] and Drew et al.[31]. Briefly, the gene was cloned into a pDDGFP-2 vector under the control of a galactose inducible promotor (GAL1) to produce a *C*-terminal GFP fusion protein. After yeast transformation, positive expression clones were screened by whole-cell and in-gel fluorescence. The gene sequence of the *RiGT2* and the mutant were confirmed by sequencing (Eurofin Genomic) after DNA extraction (GeneJET Plasmid Miniprep Kit, Thermo Fisher Scientific).

### Preparation of alcohol-insoluble residues (AIRs) of *R. ilealis* CRIB[T] and LoGSA yeast cells
The cultivation conditions of *R. ilealis* CRIB[T] are described in the instructions from DSMZ, and conditions for the cultivation of LoGSA cells are described in the next section. After centrifuging (3000×*g*,

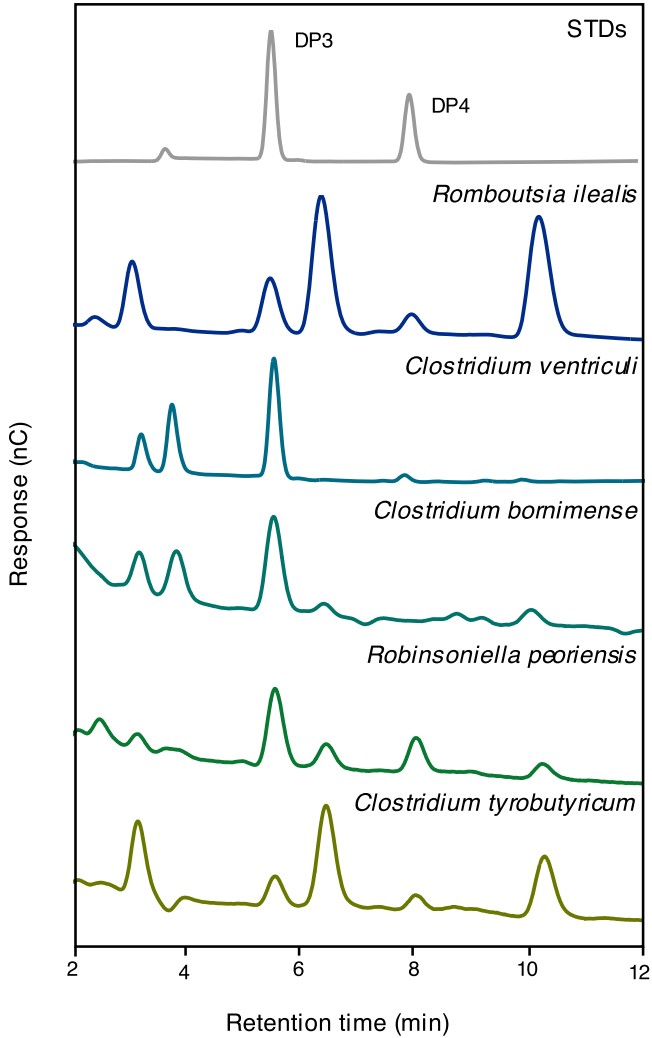

**Fig. 9 | (1,3;1,4)-β-ᴅ-Glucan oligosaccharide profiles of selected Gram-positive bacteria.** EPSs from *C. ventriculi, C. bornimence, R. peoriensis,* and *C. tyrobutyrisum* were digested by lichenase and the hydrolysates analyzed by HPAEC-PAD and compared with a similar experiment with the EPS from *R. ilealis.* A mixture of standard pure DP3 and DP4 oligosaccharides was also analyzed under the same conditions. Peaks with the same retention time as the standard DP3 are present in the hydrolysates of all species and peaks with the same retention time as the standard DP4 are present in the hydrolysates of all species except *C. bornimence.* The chromatogram with control groups aligned is shown in Supplementary Fig. 17. STDs, oligosaccharide standards. DP degrees of polymerization.

10 min), the cells were collected and resuspended in 75% (v/v) aqueous ethanol solution, followed by sonication (Branson, Brookfield, USA) (40% amplitude, 30 s, 3 min total) for *R. ilealis* CRIB[T] and using a French Press (1000 psi; two rounds) for the yeast cells. The broken cells were washed with 100% ethanol and 100% acetone and collected by centrifuging (3000×*g*, 10 min) at room temperature (RT). After the washed AIRs had been completely air-dried, they were resuspended in deionized water, transferred to dialysis tubes (Cat. No. D0530-100FT, Sigma), and dialyzed with continuously running Milli-Q water for 16 h and freeze-dried.

### Whole-cell and in-gel fluorescence
Selected LoGSA colonies were inoculated into 5 mL of -URA medium (2 g/L of yeast synthetic drop-out medium without uracil, and 6.7 g/L yeast nitrogen base without amino acids) containing 2% glucose, and were incubated overnight on an orbital shaker at 30 °C at 280 rpm. The overnight cultures were adjusted to $OD_{600}$ 0.12 in a total of 10 mL of

-URA medium containing 0.1% glucose in 50 mL centrifuge tubes (Corning™ Mini Bioreactor Centrifuge Tube). The cultures were incubated at 30 °C at 280 rpm and induced with 20% galactose to make a final concentration of 2% galactose when the $OD_{600}$ reached 0.6. The induced cultures were collected after 22 h and centrifuged (3000×*g*, 5 min) at 4 °C. The pelleted cells were resuspended in 200 µL of YSB buffer (50 mM Tris–HCl pH 7.6, 5 mM EDTA, 10% glycerol) and transferred to a black Nunc 96-well optical bottom plate. The whole-cell fluorescence emission was measured at 512 nm by excitation at 488 nm in a microplate spectrofluorometer (Clariostar, BMG Labtech). The clones with successful expressions were collected and transferred to Eppendorf tubes. The cells were broken by glass beads in a Tissue lyzer (Retsch GmbH, Haan, Germany) at 30 Hz for 7 min at 4 °C and pelleted by centrifugation (3000×*g*, 5 min). The supernatants were collected and centrifuged (17,000×*g*, 1 h), and the crude membranes were pelleted and mixed with sample loading buffer (Invitrogen). After incubation for 10 min at 37 °C, the mixtures were transferred into SDS–PAGE (Bio-Rad), and electrophoresis was carried out at 150 mV. The gel was analyzed with a CCD camera system (Las 1000 gel system, Fujifilm). All the steps were conducted at 4 °C unless indicated otherwise.

### Neutral sugar linkage analysis of the oligosaccharides and polysaccharides
Linkage analyses of the polysaccharides and the oligosaccharides were conducted as previously reported[67] with slight modifications. Briefly, the purified oligosaccharides (-1 mg) from bacteria and genetically modified yeast were reduced by NaBD₄ (5 mg) in 0.5 mL of deionized water with magnetic stirring overnight, followed by the quenching of excess reductant with acetic acid, evaporating the resulting solution to dryness under nitrogen, and the repeated evaporation to dryness in 0.5 mL of 10% (v/v) acetic acid in methanol. The resulting oligosaccharide alditols were converted into their partially methylated alditol acetate (PMAA) derivatives by permethylation with methyl iodide in dimethyl sulfoxide and sodium hydroxide[68], hydrolysis with 4 M trifluoroacetic acid (TFA) at 100 °C for 2 h, reduction with NaBD₄[69], and peracetylation in a heated mixture of acetic anhydride/TFA (5:1, v/v)[70]. *R. ilealis* CRIB[T] AIRs were treated with α-amylase from porcine pancreas (Sigma-Aldrich®, USA) followed by extensive dialysis against deionized water and freeze-drying. The preparation of PMAAs from the α-amylase-treated polysaccharides was then conducted without treatment with NaBD₄ before the methylation step. The PMAAs from the oligosaccharides and polysaccharides were analyzed on an Agilent 7890A-5977B GC-MS system (Agilent technologies, USA) with a Supelco SP2380 capillary GC column (100 m × 0.25 mm × 0.20 µm) with a constant column outlet helium flow rate of 1.2 mL/min and with an initial oven temperature of 140 °C, which was increased at 1.25 °C/min to 190 °C, and then at 8 °C/min to 250 °C (held for 30 min). Data were collected and analyzed by Agilent OpenLab CDS (version 2.6) software. The separated PMAAs were identified by their relative retention times and by comparing their MS fragmentation patterns with those of reference derivatives and by referring to the literature[71]. Two separate experiments were conducted for each sample.

### Treatment with lichenase and (1,3;1,4)-β-ᴅ-glucan oligosaccharide purification
AIRs (2 mg) were resuspended in 200 µL of 20 mM phosphate buffer (pH 6.5) and treated with 50 µL of (1,3;1,4)-β-ᴅ-glucanase (lichenase) from *Bacillus subtilis* (Megazyme International Ireland Ltd)[72]. After 24 h of incubation at 50 °C, the mixtures were centrifuged (3000×*g*, 5 min), the supernatants loaded onto Bond Elut Carbon cartridges (Agilent Technologies, USA), and washed with $H_2O$. The (1,3;1,4)-β-ᴅ-glucan

oligosaccharides were eluted with 50% CH$_3$CN and freeze-dried for subsequent analysis.

## MALDI-TOF-MS

Samples were prepared by mixing 10 µL of the lichenase digest with 10 µL of 2,5-dihydroxybenzoic acid (DHB) [10 mg/L in 0.1% (v/v) TFA and 50% (v/v) CH$_3$CN] and 6 µL of NaCl solution (10 mM). The mixtures were spotted onto a steel plate and air-dried. MALDI-TOF MS was performed with an Applied Biosystems 4800 MALDI instrument in the linear positive-ion mode. The crystalized analytes were ionized with an N$_2$-laser and accelerated into a time-of-flight analyzer. The number of shots collected for the resulting spectra was determined by the response from the analytes.

## High-performance anion exchange chromatography with pulsed amperometric detection (HPAEC-PAD)

The oligosaccharides were also analyzed by high-performance anion exchange chromatography with pulsed amperometric detection (HPAEC-PAD) using an ICS-3000 system (Dionex, Sunnyvale, CA, USA) and a CarboPac-PA1 analytical column (4 × 250 mm, Thermo Scientific, USA) fitted with a CarboPac PA1 guard column (4 × 50 mm, Thermo Scientific, USA). Before samples were injected, the column was washed with 200 mM NaOH for 10 min at a flow rate of 1 mL min$^{-1}$ and equilibrated for 5 min with 100 mM NaOH and 5% (v/v) 1 M NaOAc. The samples were eluted with a constant 100 mM NaOH solution and a gradient of 1 M NaOAc from 5% to 25% over 25 min. Pure DP3 (G4G3G, M3) and DP4 (G4G4G3G, M4) (1,3;1,4)-β-D-glucan oligosaccharide standards, and pure oligosaccharide standards derived from laminarin (laminaribiose L2, laminaritriose L3, and laminaritetraose L4) and cellulose (cellotriose C3, cellotetraose C4, cellopentaose C5, and cellohexaose C6) were obtained from Megazyme.

## Cultivation of selected anaerobic bacteria from class Clostridia

The medium used to culture all species in this study was prepared from the recipe on the DSMZ GmbH website (https://www.dsmz.de). Before opening the vials of bacteria (DSM 286 *Clostridium ventriculi* (syn. *Sarcina ventriculi*), DSM 2637 *Clostridium tyrobutyricum*, DSM 25664 *Clostridium bornimense*, DSM 102218 *Clostridium nigeriense*, DSM 100592 *Niameybacter massiliensis*, and DSM 106044 *Robinsoniella peoriensis*), medium was injected into 50 mL serum bottles, which were covered tightly with rubber to prevent gas leakage. After autoclaving, all the media were purged with 100% N$_2$ to ensure an anaerobic environment. When the media were all prepared, the vials containing the bacteria were opened and inoculated in the anaerobic chamber. The inoculated media were purged with 100% N$_2$ again and cultured at 37 °C for 3–5 days on an orbital shaker at 100 rpm. All the inoculation procedures followed the instructions from DSMZ GmbH.

## Preparation of microsomal fractions (MFs) and *Ri*GT2-GFP membrane protein purification

The selected LoGSA colonies were inoculated into 5 mL of -URA medium containing 2% glucose and were incubated overnight on an orbital shaker at 30 °C at 280 rpm. The overnight cultures were adjusted to OD$_{600}$ 0.12 in 1 L of -URA medium containing 0.1% (w/v) glucose in a 5 L flask. The cultures were incubated at 30 °C at 280 rpm, and induced with 20% (w/v) galactose to make a final concentration of 2% (w/v) galactose when the OD$_{600}$ reached 0.6. The induced cultures were collected after 22 h and centrifuged (3000×$g$, 5 min, 4 °C). The cells were resuspended in a buffer (pH 7.4) containing 50 mM HEPES, 150 mM NaCl, 10% glycerol, and 1 tablet of cOmplete™ Protease Inhibitor Cocktail (Roche, Basel, Switzerland). The cell suspensions were then passed twice through a French Press (1000 psi) (SLM Aminco), and the cell debris was removed by centrifugation (3000×$g$, 5 min,

4 °C). The microsomal fractions (MFs) were collected by centrifugation (150,000×$g$, 1 h) in a Beckman Ti70 rotor, before being resuspended in 100 mM MOPS buffer (pH 7) containing 10% glycerol. The protein concentrations of the MFs were adjusted to 1.5 mg/ml for the radiometric in vitro assays. For the protein purification, the cultivation protocol was similar, but with some steps modified for the increase in scale: the overnight cultures were adjusted to an OD$_{600}$ of 0.12 in 8 L of -URA medium containing 0.3% (w/v) glucose and incubated for 20 h before induction with 20% (w/v) galactose, with a final concentration of 2% (w/v) galactose. For the *Ri*GT2-GFP protein solubilization, the pelleted MFs were solubilized in a buffer containing 50 mM HEPES (pH 7.4), 150 mM NaCl, 10 mM MgCl$_2$ and 10% (v/v) glycerol, 20 mM imidazole and 1% (w/v) *n*-dodecyl-β-D-maltoside (DDM)/0.01% (w/v) cholesteryl hemisuccinate (CHS) for 1 h at 4 °C with gentle stirring. The insoluble material was separated by centrifugation (150,000×$g$, 1 h) in a Beckman Ti70 rotor. The soluble membrane fraction was then incubated with Ni-NTA His Bind Resin (Millipore, USA) at 4 °C for 1 h with gentle stirring. The resin was packed in a gravity flow chromatography column, washed with a wash buffer containing 50 mM HEPES (pH 7.4), 150 mM NaCl, 10 mM MgCl$_2$ and 10% glycerol, 50 mM imidazole and 0.02% (w/v) DDM/0.002% (w/v) CHS in a 5-column volume, and eluted by an elution buffer containing 50 mM HEPES (pH 7.4), 150 mM NaCl, 10 mM MgCl$_2$ and 10% (v/v) glycerol, 500 mM imidazole and 0.02% DDM/0.002%CHS. The eluted proteins were then separated by SDS–PAGE, and the identifications of the proteins were confirmed by in-gel tryptic digestion followed LC–MS.

## Radiometric in vitro assay

A total of 100 µL MFs (1.5 mg/mL) from transgenic or non-transgenic LoGSA cells was added to a 100 µL solution containing 100 mM MOPS (pH 7), 18 mM MgCl$_2$, 4 mM UDP-glucose, 0.2 µCi/ml UDP-[$^{14}$C]-glucose. The c-di-GMP was added at the indicated concentrations, the reactions were incubated at RT for 1 h, and 95% (v/v) ethanol (400 µL) was added to terminate the reaction. The mixtures were then passed through Whatman® glass microfiber filters (binder-free). The remaining insoluble products on the filter were washed with ddH$_2$O and 66% (v/v) ethanol, and then dried. The filter was submerged in Ultima Gold F scintillation reagent (PerkinElmer, Waltham, MA, USA) before being scintillation counted. All measurements were performed in triplicate. Error bars represent standard deviation (SD).

## Surface plasmon resonance (SPR)

The *Ri*PilZ-GFP was expressed in LoGSA yeast as indicated above and the harvested cells resuspended in PBS buffer (pH 7.4) with 1 tablet of cOmplete™ Protease Inhibitor Cocktail and disrupted by twice passing through a French Press (1000 psi). The cell debris was removed by centrifugation (3000×$g$, 5 min, 4 °C) and the supernatant was incubated with Ni-NTA His-Bind Resin (Millipore, USA) for 1 h at 4 °C with gentle stirring. The resin was packed in a gravity flow chromatography column, washed with PBS (pH 7.4) containing 10% glycerol and 50 mM imidazole in a 5-column volume, and eluted with PBS (pH 7.4) containing10% (w/v) glycerol and 500 mM imidazole. The eluted protein was then concentrated, and the buffer was switched to PBS (pH 7.4) to remove imidazole using Amicon Ultra-0.5 mL centrifugal filters (Millipore, USA). The ability of c-di-GMP to bind to *Ri*PilZ-GFP fusion protein was determined using a Biacore T200 surface plasmon resonance instrument. *Ri*PilZ-GFP was applied with sodium acetate buffer (pH 5.0) and immobilized to a level of about 12,000 response units on a CM5 sensor chip with an amine coupling kit (Cytiva, Sweden). Then, a two-fold serially diluted analyte, c-di-GMP, starting at a concentration of 40 µM was injected into the flow channels in PBS buffer. Each condition was implemented at a flow rate of 10 µL/min for 60 s at 25 °C. The signals obtained were subtracted from the reference channel that had not been coated with *Ri*PilZ-GFP and plotted in a resonance unit against a time sensorgram.

## Isothermal titration calorimetry (ITC)

The *Ri*PilZ-GFP was expressed and purified as indicated above. Measurements were carried out at 25 °C in a MicroCal ITC200 system (MicroCal, Northampton, MA), with 40 μM *Ri*PilZ-GFP in the cell and 400 μM c-di-GMP in the syringe. Samples (16) 2.5 μL in volume were injected at 1 min intervals with stirring at 1000 r.p.m. The data were fitted with Microcal PEAQ-ITC analysis software supplied by the manufacturer.

## Indirect immunofluorescence microscopy using an inverted confocal microscope

Yeast cells from 10 mL cultures were centrifuged (4000×*g*, 10 min, 4 °C) and resuspended in PBS buffer (pH 7.4) resulting in a final volume of 300 μL. To prepare spheroplast cells, 30 μL of 10 mg/mL of Lyticase (from *Arthrobacter luteus*, Sigma-Aldrich®) in PBS and of β-mercaptoethanol (6 μL) were added and incubated on an orbital shaker at 30 °C at 200 rpm for 2 h. The spheroplast cells were collected by centrifugation (4000×*g*, 5 min, RT) and washed twice with PBS. The cells were then fixed with 4% (w/v) paraformaldehyde (PFA) with shaking at 200 rpm for 75 min at 30 °C. The fixed cells were washed with PBS and incubated in 0.1% (w/v) bovine serum albumin (BSA) in PBS for 15 min in RT and harvested by centrifugation (4000×*g*, 3 min, 4 °C). The cells were then resuspended in a solution containing the primary antibody, a mouse monoclonal IgG antibody against (1,3;1,4)-β-ᴅ-glucan (Biosupplies Australia Pty Ltd) (BS400-3) diluted 1:100 with 0.1% (w/v) BSA in PBS and incubated at 4 °C overnight. The cells were then washed with PBS (4 times) by centrifuging (4000×*g*, 3 min, 4 °C) and resuspended in the secondary antibody, goat anti-mouse IgG (H + L) labeled with the fluorophore Alexa Fluor™ 555, (Superclonal™ Recombinant Secondary Antibody, Alexa Fluor™ 555) (ThermoFisher), diluted 1:100 with 0.1% (w/v) BSA in PBS and incubated for 1 h at RT in the dark. The cells were washed after the incubation with the primary antibody, and then pipetted onto microscope slides and mixed with 1 μg/mL DAPI (a fluorescent DNA stain) in SlowFade™ antifade mountant (Thermo Fisher). The slides were incubated in the dark for at least 5 min and then examined using an inverted confocal microscope (Leica TCS SP5). Leica microsystem LAS AF software was used to view as well as export images of cells, and ImageJ and Fiji were used to analyze and generate images. The lasers used were as follows: diode 405 and 461 nm (blue); Ar 488, 518 nm (green); and DPSS (diode-pumped solid-state) 561 and 565 nm (red).

## Indirect immunogold microscopy using transmission electron microscopy (TEM)

**Embedding and sectioning.** The yeast cells were briefly fixed in 8% (w/v) formaldehyde in 0.1 M phosphate buffer (pH 7.4), centrifuged (3000×*g*, 1 min), and replaced with a mixture of 5% (v/v) glutaraldehyde and 3% (w/v) paraformaldehyde in the same buffer for 1 h at RT, followed by washing in this buffer. To permeabilize the yeast cell walls, the samples were treated with 1% (w/v) sodium metaperiodate for 10 min. This was followed by extensive washing in phosphate buffer, dehydration in a graded series of concentrations of aqueous ethanol (30–80%), and infiltration with LR White Resin (Ted Pella, CA, USA). The resin was heat polymerized (55 °C, 48 h) the following day and ultrathin sections (~60 nm thick) were cut on an ultramicrotome (Leica EM UC7 Wetzlar, Germany) and placed on formvar-coated mesh grids.

**Immunogold labeling.** Grids were incubated with 50 mM glycine for 15 min followed by Aurion Blocking Solution (Aurion, Wageningen, Netherlands) for 30 min. After further incubation with BSA-C ™ (0.15%) (Aurion) for 15 min, the grids were placed on a drop of the mouse monoclonal antibody against (1,3;1,4)-β-ᴅ-glucan (5 μg/mL; BS400-3, Biosupplies Australia Pty Ltd) in BSA-C™ (0.15%) or only BSA-C™ for 1 h. The grids were washed in BSA-C ™ and then incubated (1 h) with a goat anti-mouse secondary antibody conjugated to 12 nm colloidal gold particles (1:100, Jackson ImmunoResearch, PA, USA). After extensive washing in BSA-C ™ and Milli-Q water, the grids were stained with 5% (w/v) uranyl acetate and Reynold's lead citrate. The grids were examined using a Tecnai G2 Spirit BioTWIN transmission electron microscope (Tecnai/Thermofisher) operated at 80 kV and images were obtained with an Orius SC200 CCD camera (Gatan Inc/Blue-Scientific).

## Theoretical modeling of the *Ri*TG2 structure

**Template selection.** There are several experimental structures available for GT2 enzymes, including structures of bacterial and plant cellulose synthases, as well as a structure of the (1,3;1,4)-β-ᴅ-glucan synthase CslF6 from barley. Before deciding on the best strategy for model prediction, the *Ri*GT2 sequence was compared with sequences for experimental structures using a template search in SWISS-MODEL[36]. SWISS-MODEL returned *Cereibacter sphaeroides* BcsA as the closest match with a global model quality estimate (GMQE) score of 0.65 and sequence identity of 25% (PDB 4P00[20]). The GMQE is a quality score that estimates the accuracy of a 3D model by combining properties from the target-template alignment and the template structure and ranges between 0 and 1 with a higher value indicating higher reliability. Cotton cellulose synthase had a GMQE score of 0.39 and sequence identity of 20% (PDB 7D5K[41]), while CslF6 had an even lower GMQE score of 0.36 and sequence identity of 21%. Based on these results, *Cs*BcsA is expected to be the best structural template for the overall structure of *Ri*GT2, while CslF6 could provide more accurate local structural information for binding of (1,3;1,4)-β-ᴅ-glucan.

**Model building.** The artificial intelligence software AlphaFold2[35] is superior for comparative modeling when sufficient training data exist (sequences and templates), which is the case for cellulose synthase, but tends to produce models that are biased towards the training sets. For scarce template data, such as for (1,3;1,4)-β-ᴅ-glucan synthase, traditional homology modeling using SWISS-MODEL could be useful to avoid such bias. We, therefore, decided to generate theoretical *Ri*GT2 models using both non-template-based AlphaFold2 modeling and template-based modeling with SWISS-MODEL. Modeling using AlphaFold2 was performed as implemented in the Google Colaboratory resource (ColabFold[73]; AlphaFold2.ipynb). Briefly, the input included the full-length sequence for *Ri*GT2 (UniProt A0A1V1I049), multiple sequence alignment (MSA) mode MMseqs2 (Many-against-Many sequence searching), unpaired_paired as pair mode (pair sequences from same species and unpaired MSA), and no template information. When using SWISS-MODEL, the structure of *Hv*CslF6 was selected as a template (UniProt B1P2T4; PDB 8DQK[37]).

**Alignment of the sequences for CsBcsA and *Ri*TG2.** The sequence of *Ri*GT2 (UniProt A0A1V1I049) was aligned with that of the closest homolog with known experimental 3D structure, i.e., the catalytic subunit of *C. sphaeroides* cellulose synthase (CsBcsA; UniProt Q3J125). The structure of the (1,3;1,4)-β-ᴅ-glucan synthase CslF6 from barley (*Hv*CslF6; UniProt B1P2T4; PDB 8DQK[37]) has been determined experimentally but has overall lower sequence similarity to *Ri*TG2 and *Cs*BcsA and was therefore not included in the alignment. An initial pre-alignment was generated using Clustal Omega[74] (https://www.ebi.ac.uk/Tools/msa/clustalo/). The theoretical model of *Ri*TG2 was then superposed with the experimental crystal structure of *Cs*BcsA (PDB 4HG6[19]) using SSM Superpose in Coot[75,76] to produce an optimal three-dimensional match of the two coordinate sets. The pre-alignment was then manually optimized to match with the structural superposition. The final alignment was visualized with ESPript3.0[66] (https://espript.ibcp.fr/ESPript/ESPript/).

**Reporting summary**

Further information on research design is available in the Nature Portfolio Reporting Summary linked to this article.

## Data availability

The data that support the findings of this study are available within the article and its Supplementary Information. Source data are provided with this paper.

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

## Acknowledgements

This work was supported by the Knut and Alice Wallenberg Foundation, the KTH Royal Institute of Technology, and the National Science and Technology Council, Taiwan (MOST, 110-2636-M-038-001). We would like to thank Dr. Ann C.Y. Wong and Prof. Philip J Harris for critical reading and further assistance with the manuscript; Dr. David Drew for the pDDGFP-2 plasmid; Dr. Stefan Klinter for LoGSA yeast strain; Dr. Johan Larsbrink for expert advice and helpful discussion; Dr. Sheng-Wei Lin for assistance with the SPR; Chun Austin Changou,

Huei-Min Chen, and the Taipei Medical University Core Facility Center for assistance with the confocal and electron microscopy; Dr. Monica Hodik and the BioVis platform of Uppsala University for assistance with the electron microscopy; Dr. Dayanand Kalyani for advice with protein purification; Dr. Lauren McKee for advice on radiometric in vitro assay, Dr. Vaibhav Srivastava for assistance with the identification of tryptic peptides.

## Author contributions

S.-C.C., Y.S.Y.H. conceived and designed the project. S.-C.C., M.-R.K., R.K.S., S.M.D.-M., X.X., J.Y., V.F., C.D. conducted the experiments. S.-C.C., R.K.S., X.X., F.V., D.W.A., V.F., C.D., and Y.S.Y.H. analyzed the data. S.C., R.K.S., X.X., V.F., C.D., and Y.S.Y.H. wrote the manuscript. All authors read and approved the final manuscript.

## Funding

## Competing interests

The authors declare no competing interests.
