## [Peer Review File · Nature Communications]

REVIEWER COMMENTS

Reviewer #1 (Remarks to the Author):

Authors identified and characterized a bacterial GT2 enzyme that synthesizes mixed-linkage glucan in *Romboutsia ilealis* CRIB, a Gram-positive bacterium. By biochemical approaches, authors demonstrated that *R. ilealis* produces MLG consisting of DP3 and DP4 units. To verify the GT2 enzyme in *R. ilealis*, they expressed the candidate GT2 in yeast, detected the MLG and verified the product by detailed analysis. Finally, they tested other Gram-positive bacteria and presented that other Gram-positive bacteria also have MLG. This work presented the result in a simple and clear way to follow, but I have a couple of questions and suggestions to improve the manuscript.

Major points

1. PilZ domain. Authors tested whether PilZ domain can regulate MLG biosynthesis by point mutation. However, whether the point mutation is effective was not verified in this work. The mutation needs to be verified by, at least, the direct binding assay as shown in the reference 5. Also, authors also need to present the level of protein expressed in RiGT2 and RiGT2:R517A transgenic yeast to determine the difference of MLG production.
2. MLG in Gram-positive bacteria
 - a. I think authors need to explain in more detail. Why did you choose a subset of bacteria? Do they have a phylogenetic connection?
 - b. Figure S6 shows the sequence alignment of RiGT2 homologs. Are they from the bacteria only with MLG in the EPSs? Did you find some motifs conserved in the RiGT2 homologs producing MLG compared to GT2 homologs not producing MLG? I think you can also color the sequence based on the MLG result. Then, please discuss possible explanations (conserved domain? addition of c-di-GMP?).
 - c. To ensure the experimental condition, do you have any marker to present the bacteria without MLG secrete EPS?

Minor points

1. Page 5 line 118. What fractions did you test? You need to clarify fractions.
2. Fig S1. The 4-linked Glc : 3-linked Glc ratio looks higher than 4:1
3. Page6 Line 120-121. What is the DP3:DP4 ratio in Fig 1? Does 4-linked Glc : 3-linked Glc ratio clearly support the DP3:DP4 ratio in *R. ilealis* CRIBT EPSs ? According to the paper you cited, wheat has 3-4.5:1 (DP3/DP4) and 2.2-2.6:1 (1,4 : 1,3).
4. Fig 5 and S4. You can consider adding the image of tri-saccharide showing 1,4 and 1,3 linkages and indicate each sugar in the fig 5c.
5. Fig S4 legend. 3-Glcp -> t-Glcp
6. Fig S5. Error bar is missing. Statistic analysis was not explained. What's the unit? "Mass of produced MLG (mg) per ?"
7. Page 10 line 220, "mobilization of endosperms" is not clear. What do you mean?

Reviewer #2 (Remarks to the Author):

Remarks to the Author:

The study, entitled "A novel bacterial GT2 polysaccharide synthase can produce mixed-linkage (1,3;1,4)- β -glucans" (Yves S.Y. Hsieh - corresponding author) provides insights into the synthesis of the (1,3;1,4)- β -D-glucan (mixed-linkage β -D-glucan) exopolysaccharide by *Romboutsia ilealis* CRIBT (RiGT2).

Glycoside transferases (GTs) synthesise carbohydrate polymers that are involved in a vast variety of biological processes. Understanding the structure and mechanisms of how GTs synthesise polysaccharides is important for making use of such enzymes. For example, to develop industrially important biofilms that could function as renewable biomaterials or have importance in human health.

Using molecular biology, carbohydrate chemistry, bioinformatics, a gain-of-function approach in yeasts, and other techniques, Hsieh et al., tried to develop a picture that mixed-linkage β -D-glucan exopolysaccharides are synthesised by a variety of bacteria.

Improvements are possible, including immunolabelling or immunofluorescence imaging of synthesised (1,3;1,4)- β -D-glucans at the level of yeast cells, which would significantly strengthen key concepts of this manuscript. This would require providing new data to deliver insights into the structure and mechanisms of how RiGT2 synthesises (1,3;1,4)- β -D-glucan polysaccharides.

In conclusion, I felt that the manuscript is incomplete, the statements are not properly formulated, the attention to detail is vague, the method descriptions are inadequate, and the manuscript is difficult to read. Results are fragmented, and individual data (because they are incomplete) are not supportive of each other in their conclusions convincingly, robustly, and clearly. Although the results are novel, I felt that the significance of this study is not what it could have been.

For the current form of this manuscript, a more specialised journal seems suitable oriented to molecular biology, carbohydrate chemistry, and microbiology.

Major comments:

Pg 1 (and Pg 6): "A novel bacterial GT2 polysaccharide synthase can produce mixed-linkage (1,3;1,4)- β -glucans" – The title of this manuscript is misleading. This presumably GT2 enzyme is not novel and the authors contradict themselves: "RiGT2 is classified in the Carbohydrate Active Enzymes database (CAZY; <http://www.cazy.org/>)²⁹ and the National Center for Biotechnology Information (NCBI) Database (Table S1 and S3)¹⁵."

Pg 6: The reactions of *R. ilealis* CRIBT EPS fractions (Figure 1) should have included hydrolysis with a purified endo-(1,3)- β -D-glucanase (EC 3.2.1.37). These data should have been presented to indicate that there are no consecutive (1,3)-linkages of glucose moieties in the *R. ilealis* CRIBT EPS fractions.

Pg 6: The presentation of data in Figure 2 (topology model predicted by PRED-TMR algorithm; Pasquier et al., 1999 – not properly cited of the Reference specified below) is uninformative, hard to read, and obsolete. The authors should have generated the 3D model of RiGT2 in complex with c-di-GMP to reveal its binding site (PilZ motif or domain) and other significant residues and structural elements, using current molecular modelling approaches. For example, the *Rhodobacter sphaeroides* bacterial cellulose synthase could have been used as a template for homology modelling. That would have provided proper insights into the predicted structure of RiGT2 and the visualisation of D, D, D, QxxRW are PilZ structural elements.

Reference:

Pasquier, C., Promponas, V.J., Palaivos, G.A., Hamodrakas, J.S.: A novel method for predicting transmembrane segments in proteins based on a statistical analysis of the SwissProt database: the PRED-TMR algorithm. *Protein Eng.* 12, 381–385 (1999).

Pg 7: Biochemical characteristics and identity of the expressed RiGT2 protein in yeasts are missing. These analyses need to be performed comprehensively on a partially purified protein since it has GFP and His tags. These analyses should include: (i) identity of the expressed RiGT2 protein using tryptic peptide mapping; (ii) specific activity and catalytic constants; (iii) and substrate specificity. These data will confirm that the recombinant RiGT2 protein is indeed a GT enzyme to remove doubt "We hypothesized that RiGT2 is involved in MLG biosynthesis."

Pg 7: In-gel fluorescence analysis data of the RiGT2 protein extracted from transgenic yeast indicates the molecular mass of approximately 100 kDa (Figure 3), however, this is not proof that this band corresponds to the expressed RiGT2-GFP-His8 fusion protein. First of all, the predicted molecular mass of this recombinant RiGT2 protein with C-terminal GFP

and His tags is not given. Second, as for the identity of this protein on the SDS-PAGE gel (Figure 3), it should have been removed from the SDS-PAGE gel and subjected to tryptic peptide analysis (as described above under (i)). Also, an uncropped scan of the Coomassie Brilliant Blue-stained gel of the preparations shown in Figure 3 needed to be provided (including an uncropped scan of Figure 3).

Pg 7: Typical whole-cell fluorescence data need to be provided for the reader to show the feasibility of the gain-of-function approach for RiGT2 in yeasts ("Total RiGT2 protein expression was determined by whole-cell fluorescence calibrated against a GFP standard curve; most colonies screened produced approximately 1 mg of RiGT2-GFP per liter of -URA media").

Pg 8: The existing bioinformatics component is incomplete and difficult to read. In Figure S6, it is not clear, which of the 22 presented entries are which, and the detailed annotations in the alignment are missing such, for example, the location of the PilZ motif (domain). A comprehensive phylogenetic analysis needs to be included, where selected bacterial and other (including monocot and dicot plants) CSLA, CSLF, CSLH, and CSLJ proteins need to be analysed. The robust phylogenetic tree needs to be generated, conclusions drawn and the alignment with the list of entries provided in Supplementary information. For example, the reader needs to know, if PilZ motifs (domains) also occur in the (1,3;1,4)- β -glucan synthases from grasses.

Pg 7-9: Details in the Methods section were insufficiently specified. Based on these descriptions, experiments cannot be reproduced. For example, the ultracentrifugation details (membrane fraction) are not given, media compositions of how to culture *R. ilealis* CRIBT, and many other experimental details remain unclear ("After centrifuge at 3000 x g for 10 minutes" or "centrifuged at 3000 g for 5 min" – these specifications are substandard in many ways; pH values of buffers; temperature definitions during centrifugations, concentrations of EPS preparations and lichenase reagents, etc., etc.). The information on the Leibniz Institute DSMZ (German Collection of Microorganisms and Cell Cultures GmbH) is also not given.

Other comments:

Legend to Figures 1-6, and Figures S1, S2, and S6, are uninformative, lack details, and are difficult to understand. For example, cloning if what (Figure S2). Scheme 1 was not mentioned in the context of the manuscript.

Additionally, the following statements are problematic:

Pg 5: "Interestingly, the cell-wall sugars of *R. ilealis* CRIBT has been reported to be predominantly glucose, which implies the possibility of high abundant homopolysaccharide EPSs." This statement is incorrect and confusing.

Pg 7: "We detected MLG with DP3 and a small amount of MLG with DP4." It is not clear what "small amounts" means.

Pg 7: "Comparable to the oligosaccharide profile of the EPSs isolated from *R. ilealis* CRIBT, the MLG profile from the RiGT2 transgenic yeast also has a DP3 majority but had a slight variation in DP3:DP4 ratio." It is not clear what "a slight variation" means.

Pg 8: "The detailed oligosaccharide structures of the MLGs produced had the identical chemical structure to that of *R. ilealis* CRIBT MLGs." There is a contradiction in statements between pages 7 and 8.

Pg 8: "AIRs were extracted from RiGT2 wild type (WT) and RiGT2:R517A transgenic yeasts, each with six replicas, and we found no significant differences in the amount and fine structure of MLGs synthesized (Figure S5)." Legend to Figure S5: "The result shows the average of MLG production comparison from two groups with no statistical difference." Here, standard deviations in Figure S5 are missing, and a more robust statistical analysis should have been performed.

Pg 8: G4 peaks for Robinsoniella peoriensis and Clostridium bomimense bacteria are absent in Figure 6. The profile of Clostridium ventriculi is complex and not explained, and its Reference in Figure 6a is not indicated.

Technical comments:

Acronyms (DP, GI, Hexe3) at the first use (or not at all) remain undefined.

There are inadequacies in chemistry/terminology and there is jargon usage in the manuscript throughout:

- **The term "(1,3;1,4)- β -glucan" is incorrect – correct is (1,3;1,4)- β -D-glucans (the letter D is capped)**
- **There is confusion in the usage of G4G3G, DP3 and G3**
- **"D, D, D, QxxRW are PilZ domains" – correct is residues and motifs, or structural elements; these elements cannot be collectively called domains.**
- **"key amino acid" – correct is an amino acid residue**
- **Incorrect terminology "helixes" – correct is α -helices**
- **Jargon: "Gram-positives"**
- **DP3 and DP4 "motifs" – correct is oligosaccharides**
- **And many others through the text.**

English language used throughout the manuscript is unsatisfactory, specifically:

- **Usage of articles**
- **Inappropriate usage of "their", "interestingly" (to whom?), "has been ..."**
- **Formulations of sentences are unclear in Introduction, Results, and Discussion (for example pages 4-6; 7-9; 12)**
- **And many others through the text.**

Incomplete definition (and errors in) References: 9, 11, and 34 (pages 17, 18 and 20).

Reviewer #3 (Remarks to the Author):

The manuscript by Chang et al is an interesting study of mixed linkage glucan (MLG) biosynthesis in bacteria. MLG has been reported in Sinorhizobium (as noted in the manuscript) but the biosynthesis was not elucidated, and bacteria are not reported to have genes in the same clade as the cellulose synthase like genes of plants. The manuscript by Chang et al. demonstrates that MLG is present in multiple gram positive bacteria, and prove the function of RiGT2 as an MLG synthase by heterologous expression in yeast. That an MLG synthase would be found as a GT2 enzyme with homology to cellulose synthase is not surprising, but it has also not been demonstrated before, and therefore the finding is novel.

The manuscript is generally presented clearly, and the conclusion is well supported, but I have some recommendations and concerns as listed below.

- 1. The phylogenetic context and similarity to other GT2 proteins is not clear. The alignment In Fig. 6S only shows the highly similar putative MLG synthases. But the introduction mentions bacterial BcsA and CrdS, as well as putative MLG synthase Bgs from Sinorhizobium. It would be helpful to have an alignment between selected proteins and RiGT2. An alignment could also include plant CesA and Csl proteins. It would certainly be helpful to see a phylogenetic tree that shows the distance of RiGT2 and Bgs from BcsA, and even more helpful if the tree also shows how distant these proteins are compared to the distance from plant CesA/Csl.**
- 2. The alignment in Fig. 6S is not clear. There are some 20 proteins shown, but it is not indicated from what species they are from, except for the RiGT2 protein. Maybe they are the same species as in Table S3 and Fig. 6 (?) but the order is clearly not the same. This needs to be labeled and explained better.**
- 3. Further to Fig. S6, the full proteins are not shown, i.e. they are trimmed at the N- and C-termini. That is fine, but needs to be explained better.**

4. For alignments such as Fig. 6S and the additional alignments I am recommending under point 1, they should be submitted not only in the pdf file, but also in some readable format, such as a text file in aligned Fasta format.

5. It is not clear how the RiGT2 candidate protein was found. It is mentioned on pg. 6 that the protein was identified in CAZy and NCBI. But is this the only GT2 protein in *R. ilealis*? If there are others, what do they look like? Is it possible that they are also involved in MLG biosynthesis? This needs to be explained and discussed better.

6. Ln 173 claims that the MLG produced in yeast has the same 'detailed oligosaccharide structure' as the MLG in *R. ilealis*. However, Fig. 1 shows significant DP4 in the digest from *R. ilealis*, whereas the MLG from yeast expressing RiGT2 is essentially pure DP3.

7. Fig. 1 shows a peak at 15 min in the lichenase digest. What is that? There is a smaller peak in the control. Is this a possible longer MLG digestion product with higher DP (as is also known from plants)?

8. Fig. 6 shows chromatogram from different species. It is not completely clear how the lichenase digests were done. More importantly, there should be controls without lichenase treatment.

**I have not attempted to comment on minor typos etc, but I will note that especially the Discussion has several odd sentences that are either typos or just poor English. Examples:
Ln 221: 'MLG has been found in the cell walls and indicated as a major architectural role in supporting cell structure'**

Ln 225: 'The origin of *R. ilealis* CRIBT in GI tract could insinuate its MLGs could possibly facilitate its adhesion to the intestinal surface'

Ln 206: 'This suggests that the MLG synthase in *R. ilealis* CRIBT reported in this study are structurally more closely related to lichenin than to the Gram-negative bacteria *S. meliloti*.'

RESPONSES TO REVIEWERS COMMENTS

Reviewer #1

Authors identified and characterized a bacterial GT2 enzyme that synthesizes mixed-linkage glucan in *Romboutsia ilealis* CRIB, a Gram-positive bacterium. By biochemical approaches, authors demonstrated that *R. ilealis* produces MLG consisting of DP3 and DP4 units. To verify the GT2 enzyme in *R. ilealis*, they expressed the candidate GT2 in yeast, detected the MLG and verified the product by detailed analysis. Finally, they tested other Gram-positive bacteria and presented that other Gram-positive bacteria also have MLG. This work presented the result in a simple and clear way to follow, but I have a couple of questions and suggestions to improve the manuscript.

1. PilZ domain. Authors tested whether PilZ domain can regulate MLG biosynthesis by point mutation. However, whether the point mutation is effective was not verified in this work. The mutation needs to be verified by, at least, the direct binding assay as shown in the reference 5. Also, authors also need to present the level of protein expressed in RiGT2 and RiGT2:R517A transgenic yeast to determine the difference of MLG production.

Response: To further verify the c-di-GMP direct binding to the RiGT2 PilZ domain (RiPilZ), we have expressed and purified the recombinant C-terminal RiPilZ-GFP fusion protein and performed the *in vitro* binding study, using Isothermal Titration Calorimetry (ITC) (**Figure S10a**) and Surface Plasmon Resonance (SPR) (**Figure S10b**). In the binding study, no direct binding was detected between the C-terminal RiPilZ-GFP and the c-di-GMP. Because no direct binding was detected between the C-terminal RiPilZ-GFP and the c-di-GMP, the mutagenesis study became expendable. We have therefore substituted the RiGT2:R517A mutagenesis result originally shown in Figure S5 with the new **Figure 6**, in which the radiometric assay shows that the RiGT2 activity remains high with or without c-di-GMP, demonstrating that the RiGT2 microsomal fraction can mediate *in vitro* MLG synthesis in the absence of c-di-GMP.

In some polysaccharide synthases, for example, the cellulose synthase (BcsA) of *Rhodobacter sphaeroides*, the PilZ domain composed of RxxxR and D/NxSxxG c-di-GMP binding motifs has been confirmed as a c-di-GMP binding domain. The PilZ domain of the MLG synthase of *R. ilealis* is not an exception and also contains the two binding motifs. Conformation differences between the interactions between c-di-GMP and the PilZ domains of BcsA and RiGT2 were verified by the structural alignment of the resolved crystal structure of c-di-GMP bound to BcsA (PDB: 5ej1) and the RiGT2 model built on AlphaFold (**Figure S11**). The RiGT2 PilZ has a smaller c-di-GMP binding pocket than the PilZ of BcsA. The Arg of the RiGT2:R582 side chain points inwards to the binding pocket, possibly interfering with the binding of dimerized c-di-GMP.

To further verify that the PilZ domain is necessary for RiGT2 activity, we generated a RiGT2 with the PilZ domain deleted (*RiGT2-PilZD*) and transformed the gene into LoSGSA-2 yeast. Interestingly, after lichenase digestion of AIR prepared from the transgenic yeast expressing C-terminal truncated RiGT2, we found no MLG-derived oligosaccharides and therefore no MLG was synthesized (**Figure S12a**). This result was further supported by the radiometric *in vitro* assay (**Figure S12c**): no activity was detected when the PilZ domain had been cropped from the RiGT2. Despite no interaction between the RiGT2 PilZ domain and the secondary messenger c-di-GMP, the presence of the C-terminal PilZ domain remains crucial for the activity of MLG synthase, possibly by facilitating the insertion and assembly of the native conformation of membrane-anchored RiGT2.

As indicated above, we have added new figures and also new text.

2. MLG in Gram-positive bacteria

a. I think the authors need to explain in more detail. Why did you choose a subset of bacteria? Do they have a phylogenetic connection?

Response: The subset of bacteria (*Clostridium bornimense*, *Clostridium ventriculi*, *Robinsoniella peoriensis* and *Clostridium tyrobutyricum*) was chosen based on 1) they have RiGT2 homologs with high sequence similarities. 2) The bacterial species are commercially available from DSMZ. 3) Out of six species that are available, four species *C. bornimense*, *C. ventriculi*, *R. peoriensis* and *C. tyrobutyricum* are culturable in our anaerobic system, while others (*Clostridium nigeriense* and *Niameybacter massiliensis*) grow only very slowly or have no growth and we were unable to harvest enough EPSs for the subsequent analysis. All bacteria with identified RiGT2 homologs with high sequence homology are positioned by phylogeny within the same order Eubacteriales of the class Clostridia. We have added a paragraph in the text to address this comment.

b. Figure S6 shows the sequence alignment of RiGT2 homologs. Are they from the bacteria only with MLG in the EPSs? Did you find some motifs conserved in the RiGT2 homologs producing MLG compared to GT2 homologs not producing MLG? I think you can also color the sequence based on the MLG result. Then, please discuss possible explanations (conserved domain? addition of c-di-GMP?).

Response: In the original Figure S6, a total of 22 putative RiGT2 homologs were identified. Of these, 11 putative RiGT2 homologs were derived from the four bacterial species *C. bornimense*, *C. ventriculi*, *R. peoriensis*, and *C. tyrobutyricum*. There is little information about the composition of the EPSs of these bacteria, but we believe that the EPSs contain more than just MLGs, possibly including cellulose.

Before addressing the putative conserved motif in MLG synthase, we would like to clarify that we have carried out a more comprehensive phylogenetic analysis as suggested by Reviewer 3. We have identified a total of 43 RiGT2 homologs. The new tree is now shown in **Figure S13**. Because of the large file size, we can provide sequence coverage and identical percentage only upon request. Specifically, a comparison of Gram-positive bacterial MLG synthases and cellulose synthases by aligning five Cgs and four CcsA. The Cgs are putative MLG synthases closely related to RiGT2, and the CcsA are cellulose synthases, putatively determined in the literature, but have not been biochemically characterized. The protein alignment shows three possible conserved sequences in Cgs that are marked in the red square (**Figure S14a**). We can only speculate that three of the marked regions could be bacterial MLG conserved motifs because this is the first Gram-positive bacterial MLG synthase that has been biochemically characterized. Another conserved motif worthy of note is Interface helices 3 (IF3). A recent study showed that IF3 in barley MLG synthase (CsIF6) affects the 1,3 to 1,4 ratios of MLGs. Further research is needed to determine if IF3 in the bacterial MLG synthase acts in a similar way.

c. To ensure the experimental condition, do you have any marker to present the bacteria without MLG secrete EPS?

Response: Thank you. A key marker that is missing and which should serve as a control is the bacterial EPS only without lichenase treatment. We have now added this control (see **Figure S15**).

Minor points

1. Page 5 line 118. What fractions did you test? You need to clarify fractions.

Response: We apologize for the confusion. We obtained the EPS (including cell walls) of *R. ilealis* CRIB by preparing an alcohol-insoluble residue. The sample for the linkage analysis was not fractionated. We have deleted the word “fraction”.

2. Fig S1. The 4-linked Glc : 3-linked Glc ratio looks higher than 4:1

Response: Fig S1 shows the TIC chromatogram of the PMAAs representing the neutral sugar linkages of the EPS of *R. ilealis* CRIB. The ratio of 4-Glcp to 3-Glcp is much higher than 4:1. This is because the whole EPS preparation may contain other polysaccharides containing 4-Glcp, such as cellulose. The linkage analysis confirms the existence of 3-Glcp and 4-Glcp linkages in the EPSs, but we cannot disprove that the 3-Glcp is derived from a 3-linked glucan (e.g. pachyman) and 4-Glcp is from a 4-linked glucan (e.g. cellulose). To confirm the existence of MLG in the EPS, as we have done in the manuscript, we treated the bacterial EPS preparation with lichenase, an enzyme that specifically cleaves MLG (generating a series of mixed linkage oligosaccharides with their reducing ends being 3-linked and the other residues 4-linked), but it cannot cleave polysaccharides such as the pachyman or cellulose. The presence of MLG can be confirmed by analysing the oligosaccharides released by lichenase by HPAEC-PAD, using standard barley MLG-derived oligosaccharides (M3 and M4) with known structures as reference compounds (**Figure 1**).

3. Page6 Line 120-121. What is the DP3:DP4 ratio in Fig 1? Does 4-linked Glc : 3-linked Glc ratio clearly support the DP3:DP4 ratio in *R. ilealis* CRIBT EPSs? According to the paper you cited, wheat has 3-4.5:1 (DP3/DP4) and 2.2-2.6:1 (1,4 : 1,3).

Response: The DP3:DP4 oligosaccharide ratio of MLG is the ratio of G4G3G:G4G4G3G. This ratio represents the fine structure of the MLG and is the measurable parameter of MLG physicochemical properties, such as solubility and aggregation tendency. The DP3:DP4 ratio can be determined by HPAEC-PAD of lichenase digested EPSs, with calibration against DP3 and DP4 standard curves. We found that the molar ratio of DP3 to DP4 in the MLG of *R. ilealis* CRIBT EPS is close to 2:1, which resembles the structure of grass MLGs. Rather surprising however, MLGs produced by *RiGT2* transgenic yeast consisted mainly of the DP3 (G4G3G) subunit. Our study suggests that the MLG produced recombinantly by *RiGT2* is structurally more closely related to lichenin (DP3 dominant) than to the MLGs of Gram-negative bacteria *S. meliloti* (DP2 only). The discrepancy of DP3:DP4 ratio could be that the recombinant *RiGT2* in the LoGSA-2 cells are not in their native lipid environment and other factors, which we have now added a small paragraph to the text in the Discussion.

The total linkage composition of EPS provides a rough estimate of MLGs and other polysaccharides that could be present. But as we have clarified earlier, the 3- and 4-Glc glucopyranosyl linkage composition of total EPS cannot be directly correlated to the linkage ratio of MLGs because some of the 3- and 4-Glcp could be derived from other EPSs.

4. Fig 5 and S4. You can consider adding the image of tri-saccharide showing 1,4 and 1,3 linkages and indicate each sugar in the fig 5c.

Response: As suggested, this image has been added to the figure as in the new **Figure 4c**.

5. Fig S4 legend. 3-Glcp -> t-Glcp

Response: Thank you. We have corrected this in new Figure S8.

6. Fig S5. Error bar is missing. Statistic analysis was not explained. What's the unit? "Mass of produced MLG (mg) per ?"

Response: The units were milligrams per liter of culture. However, as we have mentioned in our response to question 1, because no direct binding was detected between the C-terminal *RiPilZ*-GFP and the c-di-GMP, the mutagenesis study became expendable. We have substituted the *RiGT2*:R517A mutagenesis result originally in Figure S5 with the new **Figure 6**, in which the radiometric assay shows that the *RiGT2* is highly active with or without c-di-GMP, demonstrating that the *RiGT2* microsomal fraction can independently mediate *in vitro* MLG synthesis in the absence of c-di-GMP.

7. Page 10 line 220, “mobilization of endosperms” is not clear. What do you mean?

Response: Mobilization of the endosperm tissues in grasses, including cereals, is a critical process in germination. Hydrolytic enzymes, including 1,3;1,4- β -D-glucanases, degrade the polysaccharides in endosperm cell walls, and subsequently proteases and amylases degrade the storage proteins and starch within the cells.

Reviewer #2 (Remarks to the Author):

The study, entitled “A novel bacterial GT2 polysaccharide synthase can produce mixed-linkage (1,3;1,4)- β -glucans” (Yves S.Y. Hsieh - corresponding author) provides insights into the synthesis of the (1,3;1,4)- β -D-glucan (mixed-linkage β -D-glucan) exopolysaccharide by *Romboutsia ilealis* CRIBT (RiGT2).

Glycoside transferases (GTs) synthesise carbohydrate polymers that are involved in a vast variety of biological processes. Understanding the structure and mechanisms of how GTs synthesise polysaccharides is important for making use of such enzymes. For example, to develop industrially important biofilms that could function as renewable biomaterials or have importance in human health.

Using molecular biology, carbohydrate chemistry, bioinformatics, a gain-of-function approach in yeasts, and other techniques, Hsieh et al., tried to develop a picture that mixed-linkage β -D-glucan exopolysaccharides are synthesised by a variety of bacteria.

Improvements are possible, including immunolabelling or immunofluorescence imaging of synthesised (1,3;1,4)- β -D-glucans at the level of yeast cells, which would significantly strengthen key concepts of this manuscript. This would require providing new data to deliver insights into the structure and mechanisms of how RiGT2 synthesises (1,3;1,4)- β -D-glucan polysaccharides.

In conclusion, I felt that the manuscript is incomplete, the statements are not properly formulated, the attention to detail is vague, the method descriptions are inadequate, and the manuscript is difficult to read. Results are fragmented, and individual data (because they are incomplete) are not supportive of each other in their conclusions convincingly, robustly, and clearly. Although the results are novel, I felt that the significance of this study is not what it could have been. For the current form of this manuscript, a more specialised journal seems suitable oriented to molecular biology, carbohydrate chemistry, and microbiology.

Response: We have made improvements to address the concerns raised. We have carried out indirect immunofluorescence and immunogold labeling studies in which the newly synthesized MLG was detected using a monoclonal antibody specific for 1,3;1,4- β -D-glucans, see the text and **Fig. 5** and **Fig. S9**.

Major comments:

Pg 1 (and Pg 6): “A novel bacterial GT2 polysaccharide synthase can produce mixed-linkage (1,3;1,4)- β -glucans” – The title of this manuscript is misleading. This presumably GT2 enzyme is not novel, and the authors contradict themselves: “RiGT2 is classified in the Carbohydrate Active Enzymes database (CAZY; <http://www.cazy.org/>) and the National Center for Biotechnology Information (NCBI) Database (Table S1 and S3).”

Response: The gene that has been determined in this study is functionally novel. We have rephrased the title to avoid confusion.

Pg 6: The reactions of *R. ilealis* CRIBT EPS fractions (Figure 1) should have included hydrolysis with a purified endo-(1,3)- β -D-glucanase (EC 3.2.1.37). These data should have been presented to indicate that there are no consecutive (1,3)-linkages of glucose moieties in the *R. ilealis* CRIBT EPS fractions.

Response: The *R. ilealis* EPS was treated with an endo-(1,3)- β -D-glucanase as suggested; we found small peaks with the retention times of cellotriose (C3) and cellotetraose (C4), as pointed out by arrows (See Figure on the right). We found no consecutive peaks correlated with β -1,3 linked oligomers. The retention times of oligomer standards derived from cellulose, laminarins, or MLGs is shown in **Figure 1** in the main text.

Pg 6: The presentation of data in Figure 2 (topology model predicted by PRED-TMR algorithm; Pasquier et al., 1999 – not properly cited in the Reference specified below) is uninformative, hard to read, and obsolete. The authors should have generated the 3D model of RiGT2 in complex with c-di-GMP to reveal its binding site (PilZ motif or domain) and other significant residues and structural elements, using current molecular modelling approaches. For example, the Rhodobacter sphaeroides bacterial cellulose synthase could have been used as a template for homology modelling. That would have provided proper insights into the predicted structure of RiGT2 and the visualisation of D, D, D, QxxRW are PilZ structural elements.

Response: The topology model predicted by the PRED-TMR algorithm has been replaced by the AlphaFold predicted model structure (**Figure 2** and **Figure S2**), with help from Prof. Johan Larsbrink. Specifically, the *RiGT2* model has 7 TM helices, together with 3 cytosolic interface helices (IF) from the GT domain that form a tunnel containing the polymerized acceptor to incorporate incoming Glc residues. Since the *RiGT2* model resembles the structure of BcsA, and the mechanism of polymerization in BcsA has been resolved, this allows us to pinpoint several key amino acid residues in *RiGT2*. For example, the Trp from the QxxRW motif in IF2 may interact with the last residue of the acceptor polysaccharide. The Arg from xED motif catalyzes the glycosidic linkages by activating UDP-Glc (**Figure S2**). The position of the gating loop containing FxVTxK sequences may be necessary for catalytic activity. It can be regulated by the interaction between Arg from RxxxR in the PilZ domain and bacterial secondary messenger c-di-GMP.

Pg 7: Biochemical characteristics and identity of the expressed RiGT2 protein in yeasts are missing. These analyses need to be performed comprehensively on a partially purified protein since it has GFP and His tags. These analyses should include: (i) identity of the expressed RiGT2 protein using tryptic peptide mapping; (ii) specific activity and catalytic constants; (iii) and substrate specificity. These data will confirm that the recombinant RiGT2 protein is indeed a GT enzyme to remove doubt “We hypothesized that RiGT2 is involved in MLG biosynthesis.”

Response: The purified *RiGT2* recombinant protein and in-gel tryptic digestion are now added in **Figure S4b** and **Figure S6**. We have added a study of the specific activity using a radiometric *in vitro* assay of the yeast (*RiGT2*) microsomal fraction. Because other endogenous yeast enzymes from the microsomal fraction will also consume ^{14}C -labeled substrates, it is not possible to acquire reliable enzyme kinetics. However, the specific activity of *RiGT2* can be observed (**Figure 6a**). Some endogenous yeast glucan synthases are calcium-dependent (Pan et al., 2020), so the background

activity of yeast glucan synthase can be minimized by calcium depletion, increasing the signal from Mg^{2+} -dependent *RiGT2*. Based on the radioactivity of retained insoluble polymer before and after lichenase treatment, we can accept that the *RiGT2* microsomal fraction uses UDP-Glc as a substrate to produce specifically the MLG polymers.

Pg 7: In-gel fluorescence analysis data of the *RiGT2* protein extracted from transgenic yeast indicates the molecular mass of approximately 100 kDa (Figure 3); however, this is not proof that this band corresponds to the expressed *RiGT2*-GFP-His8 fusion protein. First of all, the predicted molecular mass of this recombinant *RiGT2* protein with C-terminal GFP and His tags is not given. Second, as for the identity of this protein on the SDS-PAGE gel (Figure 3), it should have been removed from the SDS-PAGE gel and subjected to tryptic peptide analysis (as described above under (i)). Also, an uncropped scan of the Coomassie Brilliant Blue-stained gel of the preparations shown in Figure 3 needed to be provided (including an uncropped scan of Figure 3).

Response: The predicted molecular weight of *RiGT2*-GFP-His fusion protein (102.7 kDa) has been added to **Figure S5**. We did not show the Coomassie Brilliant Blue-stained gel of Figure S4a (former Figure 3 in gel fluorescence) because the samples were membrane fractions from whole cell lysate for colony selection, so the *RiGT2* protein band on the stained gel could not be recognized among many others. The entire lane was like **A** shown below. Therefore, the data have been moved to **Figure S4** and zoomed out to the full lanes. We also added the purified protein gel accompanied with in-gel fluorescence in **Figure S4**. The uncropped gel and in-gel fluorescence pictures are attached below (**B** and **C**).

Pg 7: Typical whole-cell fluorescence data need to be provided for the reader to show the feasibility of the gain-of-function approach for *RiGT2* in yeasts (“Total *RiGT2* protein expression was determined by whole-cell fluorescence calibrated against a GFP standard curve; most colonies screened produced approximately 1 mg of *RiGT2*-GFP per liter of –URA media”).

Response: We have cited the reference for the calculation of whole-cell fluorescence and a table with calculated numbers was added to **Table S2**.

Pg 8: The existing bioinformatics component is incomplete and difficult to read. In Figure S6, it is not clear, which of the 22 presented entries are which, and the detailed annotations in the alignment are missing such, for example, the location of the PilZ motif (domain). A comprehensive phylogenetic analysis needs to be included, where selected bacterial and other (including monocot and dicot plants) CSLA, CSLF, CSLH, and CSLJ proteins need to be analysed. The robust phylogenetic tree needs to be generated, conclusions drawn and the alignment with the list of entries provided in Supplementary information. For example, the reader needs to know, if PilZ motifs (domains) also occur in the (1,3;1,4)- β -glucan synthases from grasses.

Response: We would like to clarify that the PilZ domain is a universal bacterial secondary messenger c-di-GMP binding domain, but it has not been identified in any plant proteins. The suggested phylogenetic analysis was performed and included cellulose synthases and MLG synthases of plant, Gram-negative, and Gram-positive bacteria (**Figure S13a**); however, because they were clearly separated from one another by the Domain or Kingdom; we felt the data were not sufficiently informative. Therefore, we have also performed a phylogenetic analysis focusing on Gram-positive bacterial MLG synthases (*RiGT2* homologs) and putative cellulose synthase CcsA **Figure S13b**. In addition to the figures, we have added text to the manuscript.

Pg 7-9: Details in the Methods section were insufficiently specified. Based on these descriptions, experiments cannot be reproduced. For example, the ultracentrifugation details (membrane fraction) are not given, media compositions of how to culture *R. ilealis* CRIBT, and many other experimental details remain unclear (“After centrifuge at 3000 x g for 10 minutes” or “centrifuged at 3000 g for 5 min” – these specifications are substandard in many ways; pH values of buffers; temperature definitions during centrifugations, concentrations of EPS preparations and lichenase reagents, etc., etc.). The information on the Leibniz Institute DSMZ (German Collection of Microorganisms and Cell Cultures GmbH) is also not given.

Response: We have now added the details requested throughout the Methods section. Text in red.

Other comments:

Legend to Figures 1-6, and Figures S1, S2, and S6, are uninformative, lack details, and are difficult to understand. For example, cloning if what (Figure S2). Scheme 1 was not mentioned in the context of the manuscript.

Response: The legend has been rewritten for clarity.

Additionally, the following statements are problematic:

Pg 5: “Interestingly, the cell-wall sugars of *R. ilealis* CRIBT has been reported to be predominantly glucose, which implies the possibility of high abundant homopolysaccharide EPSs.” This statement is incorrect and confusing.

Response: The statement has been removed.

Pg 7: “We detected MLG with DP3 and a small amount of MLG with DP4.” It is not clear what “small amounts” means.

Response: The text has been modified

Pg 7: “Comparable to the oligosaccharide profile of the EPSs isolated from *R. ilealis* CRIBT, the MLG profile from the *RiGT2* transgenic yeast also has a DP3 majority but had a slight variation in DP3:DP4 ratio.” It is not clear what “a slight variation“ means.

Response: The text has been modified.

Pg 8: “The detailed oligosaccharide structures of the MLGs produced had the identical chemical structure to that of *R. ilealis* CRIBT MLGs.” There is a contradiction in statements between pages 7 and 8.

Response: The oligosaccharides G4G3G and G4G4G3G derived from both *RiGT2* yeast MLGs and *R. ilealis* CRIBT MLGs have identical chemical structures. The statement made on page 7 was related to the DP3 (G4G3G) and DP4 (G4G4G3G) ratios. This ratio correlates to the fine structures of MLG polysaccharide and is known to influence solubility and viscosity. We hope this clarifies the confusion.

Pg 8: “AIRs were extracted from RiGT2 wild type (WT) and RiGT2:R517A transgenic yeasts, each with six replicas, and we found no significant differences in the amount and fine structure of MLGs synthesized (Figure S5).” Legend to Figure S5: “The result shows the average of MLG production comparison from two groups with no statistical difference.” Here, standard deviations in Figure S5 are missing, and a more robust statistical analysis should have been performed.

Response: Due to the comments from other reviewers, the data has been replaced. We have expressed and purified C-terminal PilZ domain-GFP fusion and tested the binding assay by ITC and SPR. Because the wildtype PilZ already showed no binding with c-di-GMP, the mutagenesis was unnecessary. Therefore, we decided to remove the RiGT2:R517A mutagenesis data and replace it by showing no difference in RiGT2 activity with and without c-di-GMP in a radiometric *in vitro* assay (See **Figure 6b**) and no binding shown in the SPR and ITC (See **Figure S10**) to prove that RiGT2 is not regulated by c-di-GMP by direct binding.

Pg 8: G4 peaks for Robinsoniella peoriensis and Clostridium bomimense bacteria are absent in Figure 6. The profile of Clostridium ventriculi is complex and not explained, and its Reference in Figure 6a is not indicated.

Response: The explanation has now been added (**Figure 8, S14**). This result indicates that MLG is more widespread in bacterial EPSs than was previously thought, and the DP3: DP4 ratio may be different between species.

Technical comments:

Acronyms (DP, GI, Hexe3) at the first use (or not at all) remain undefined.

There are inadequacies in chemistry/terminology and there is jargon usage in the manuscript throughout:

- The term “(1,3;1,4)- β -glucan” is incorrect – correct is (1,3;1,4)- β -D-glucans (the letter D is capped)
- There is confusion in the usage of G4G3G, DP3 and G3

Response: We have made the required corrections. The MLG standards G3 and G4 have been changed to M3 and M4. The usage of G4G3G, DP3, and M3 are used interchangeably for several reasons. G4G3G is used as a description of triose structure β -Glc_p-(1→4)- β -Glc_p-(1→3)- β -Glc_p. The DP3 is used largely in the description of oligosaccharide that was generated by the lichenase, especially the DP3/DP4 ratio which this ratio correlates to the fine structures of MLG polysaccharide. The M3 refers to the commercial barley G4G3G (DP3) oligosaccharide standard on the HPAEC-PAD.

- “D, D, D, QxxRW are PilZ domains” – correct is residues and motifs, or structural elements; these elements cannot be collectively called domains.
- “key amino acid” – correct is an amino acid residue
- Incorrect terminology “helixes” – correct is α -helices
- Jargon: “Gram-positives”
- DP3 and DP4 “motifs” – correct is oligosaccharides
- And many others through the text.

Response: They have been named “PilZ domain and QxxRW motifs” in previous literature as well as on the Uniprot website. Therefore, in order not to confuse readers, we have decided to keep consistency with the literature. We have made the required corrections.

English language used throughout the manuscript is unsatisfactory, specifically:

- Usage of articles
- Inappropriate usage of “their”, “interestingly” (to whom?), “has been ...”

- Formulations of sentences are unclear in Introduction, Results, and Discussion (for example pages 4-6; 7-9; 12)
 - And many others through the text.
- Incomplete definition (and errors in) References: 9, 11, and 34 (pages 17, 18 and 20).

Response: Thank you. We have made the required corrections. The English language in the manuscript has now been edited by a native writer of English.

Reviewer #3 (Remarks to the Author):

The manuscript by Chang et al is an interesting study of mixed linkage glucan (MLG) biosynthesis in bacteria. MLG has been reported in *Sinorhizobium* (as noted in the manuscript) but the biosynthesis was not elucidated, and bacteria are not reported to have genes in the same clade as the cellulose synthase like genes of plants. The manuscript by Chang et al. demonstrates that MLG is present in multiple gram positive bacteria, and prove the function of RiGT2 as an MLG synthase by heterologous expression in yeast. That an MLG synthase would be found as a GT2 enzyme with homology to cellulose synthase is not surprising, but it has also not been demonstrated before, and therefore the finding is novel.

The manuscript is generally presented clearly, and the conclusion is well supported, but I have some recommendations and concerns as listed below.

1. The phylogenetic context and similarity to other GT2 proteins is not clear. The alignment in Fig. 6S only shows the highly similar putative MLG synthases. But the introduction mentions bacterial BcsA and CrdS, as well as putative MLG synthase Bgs from *Sinorhizobium*. It would be helpful to have an alignment between selected proteins and RiGT2. An alignment could also include plant Cesa and Csl proteins. It would certainly be helpful to see a phylogenetic tree that shows the distance of RiGT2 and Bgs from BcsA, and even more helpful if the tree also shows how distant these proteins are compared to the distance from plant Cesa/Csl.

Response: We examined for possible alignment against Gram-negative bacterial BcsA, CrdS, Bgs from *Sinorhizobium* and RiGT2 as well as plant Cesa, CslF, CslH, but found only low sequence similarity; only the D,D,D,QxxRW could be aligned. Reanalysis of phylogeny (**Figure S13a**) showed cellulose (Cesa) and MLG synthases (CslF/CslH) of plants, BgsA and BcsA of Gram-negative bacteria, and CgsA (RiGT2 and homologs) and Cesa of Gram-positive bacteria are very distant and clearly separated from one another by the Domain or Kingdom; we felt the data was not sufficiently informative. Therefore, we have performed both phylogenetic analysis and protein alignment with a focus on Gram-positive bacterial MLG synthase (RiGT2 homologs), and Gram-positive bacterial putative cellulose synthase Cesa as in **Figure S13** (phylogeny) and **Figure S15** (sequence alignment).

2. The alignment in Fig. 6S is not clear. There are some 20 proteins shown, but it is not indicated from what species they are from, except for the RiGT2 protein. Maybe they are the same species as in Table S3 and Fig. 6 (?) but the order is clearly not the same. This needs to be labeled and explained better.

Response: The new alignment is now in Figure S15, with the protein listed in Table S4.

3. Further to Fig. S6, the full proteins are not shown, i.e. they are trimmed at the N- and C-termini. That is fine, but needs to be explained better.

Response: The question has been addressed. We have now provided the proteins in full length.

4. For alignments such as Fig. 6S and the additional alignments I am recommending under point 1, they should be submitted not only in the pdf file, but also in some readable format, such as a text file in aligned Fasta format.

Response: The text format of additional protein alignment has been added to **Figure S14b**.

5. It is not clear how the RiGT2 candidate protein was found. It is mentioned on pg. 6 that the protein was identified in CAZy and NCBI. But is this the only GT2 protein in *R. ilealis*? If there are others, what do they look like? Is it possible that they are also involved in MLG biosynthesis? This needs to be explained and discussed better.

Response: A table including all the GT2 enzymes in the *Romboutsia ilealis* genome has been added. There are five GT2 genes, but only one contains the typical polysaccharide synthase domains D,D,D,QxxRW. Therefore, we only evaluated this gene in our study. We have modified the text accordingly.

6. Ln 173 claims that the MLG produced in yeast has the same ‘detailed oligosaccharide structure’ as the MLG in *R. ilealis*. However, Fig. 1 shows significant DP4 in the digest from *R. ilealis*, whereas the MLG from yeast expressing RiGT2 is essentially pure DP3.

Response: Thank you for pointing this out. We have corrected the sentence. For the discrepancy, there are many other possibilities. 1) Other unknown MLG synthases in *Romboutsia ilealis* produce MLGs in different DP3:DP4 ratios. 2) Unknown proteins that could interact with MLG synthase within *R. ilealis* that switch the assembly of 3 and 4 linkages of elongating MLGs. 3) Yeast plasma membrane lipid composition is different from *Romboutsia ilealis*, which could affect the function of recombinant RiGT2. 4) C-terminal fusion GFP could also impact the function of RiGT2. We have added the text about the discrepancy observed in the Discussion.

7. Fig. 1 shows a peak at 15 min in the lichenase digest. What is that? There is a smaller peak in the control. Is this a possible longer MLG digestion product with higher DP (as is also known from plants)?

Response: These are the unknown peaks from the bacterial extract. They do not have the same retention times as β -1,4-linked, β -1,3 linked commercially available gluco-oligosaccharides (Figure 1). The same peaks also showed in the EPSs before lichenase digestion.

8. Fig. 6 shows chromatogram from different species. It is not completely clear how the lichenase digests were done. More importantly, there should be controls without lichenase treatment.

Response: We have now added control groups to the chromatogram in black in Figure S15.

I have not attempted to comment on minor typos etc, but I will note that especially the Discussion has several odd sentences that are either typos or just poor English. Examples:
Ln 221: ‘MLG has been found in the cell walls and indicated as a major architectural role in supporting cell structure’
Ln 225: ‘Its GI tract origin could insinuate the MLGs in EPS could have function of adhesion to the intestinal surface’
Ln 206: ‘This suggests *R. ilealis* CRIBT produced MLG are structurally more closer to lichenin than to the Gram-negative

Response: Thank you. We have made the required corrections. The English language in the manuscript has now been edited by a native writer of English.

REVIEWER COMMENTS

Reviewer #1 (Remarks to the Author):

The authors addressed this reviewer's questions thoroughly.

By testing the PilZ domain again, the authors provided new information about the role of PilZ in MLG synthesis different from cellulose synthase. All the new figures look good, but I want the authors to make fig 5 anti-MLG panel more visible. It is difficult to find the signals.

Reviewer #2 (Remarks to the Author):

Remarks to the Author:

(1) The title of the manuscript ("The Gram-positive bacterium *Romboutsia ilealis* has a GT polysaccharide synthase that can produce mixed-linkage (1,3;1,4)- β -D-glucans") is clearer, although it can be improved. I would advise the authors against the usage of the "GT" acronym in the title, as Nature Communication is a multidisciplinary journal.

More appropriate titles could be:

"The Gram-positive bacterium *Romboutsia ilealis* has a glycosyl transferase that can produce (1,3;1,4)- β -D-glucans"

or

"The Gram-positive bacterium *Romboutsia ilealis* has a polysaccharide synthase that can produce (1,3;1,4)- β -D-glucans"

(2) The term mixed-linkage (acronym MLG) and other derivatives (e.g. MLGs, MLG biosynthesis, MLG synthase) used in the manuscript are chemically incorrect and should be removed from the manuscript. The authors need to refer to the term "(1,3;1,4)- β -D-glucan" in all instances. There is a contradiction in the MLG term, as the MLG term could also indicate other mixed-linkage biopolymers such as e.g. (1,4;1,6)- α -D-glucans (branched α -glucans or amylopectins).

(3) Figure 2. The 3D structural models of RiGT2. The RiGT2 3D structural models were generated using AlphaFold2. RiGT2 consists of transmembrane (TM) region colored in blue, the cellulose synthase-like GT domains shown in pink, including three cytosolic interface helices (IF) colored in green and the C-terminal PilZ domain colored in orange. The gating loop is colored in pale yellow.

Currently, this figure is insufficient and doesn't explain the critical structural features of RiGT2. In this figure, the key catalytic residues (DxD, xED, QxxRW) illustrated in cpk stick representations, should have been shown, and labeled for the reader to see the catalytic machinery in the GT region (shown as GT-region, not "GT").

In the legend of Figure 2, the ranges of residues (or their numbering) of the TM-region, GT-region, and PilZ domain should have been included.

Lines 737-738 Why "models"? How many models were generated?

Lines 141-149 The text referring to Figure 2 is full of mistakes. The text should be consulted with the structural expert and the errors should be corrected (usage of domain, motif/motifs, polymerization of the β -glucan; containing the polymerized acceptor that incorporates incoming Glc residues). The terms and statements in brackets are incorrect.

Lines 346-347 "The secondary structure predictions were according to the structures from the AlphaFold2 predictions" (this sentence is incomplete).

Lines 737-741 The text of the legend is substandard and incomplete. It needs to be rewritten after the

new Figure 2 is generated. The claim of “the gating loop” is unsubstantiated and should be removed.

(4) Figure S2 AlphaFold structural models of RiGT2 show the same remarkable sequences as in BcsA. The 3D structure generated by AlphaFold2 is shown in a gold color. The c-di-GMP and polysaccharide (cellulose) structures were obtained from the resolved crystal structure of BcsA-B complex in the PDB database (5ej1) by structural comparison in UCSF Chimera. The relative position of Trp from QxxRW and Asp from xED (orange) to the acceptor polysaccharide chain are shown in the dashed square (red) with zoom-in fields in solid squares. The relative position of the first Arg (R517) from RxxxR and Asp and Ser from DxSxxG (Blue) to c-di-GMP are shown in the black squares.

General comments on Figure S2:

RiGT2 is not the structure. It is a “model” or a “structural model” based on a template (in this case PDB accession 5ej1 or any other BcsA structure). This needs to be corrected in this legend and in the entire manuscript.

The 3D structure of (what protein) generated by AlphaFold2 is shown in a gold color? How does this figure relate to Figure 2? Does it contain the model of RiGT2 (it most likely does)? So, why show it again?

Specific comments on Figure S2:

The current form of this figure is unacceptable. It is unclear, from the legend description, what structure is shown here. How was the 3D model of RiGT2 constructed; a short description as to how it was done, should have been included in the legend.

The logical approach to structural modeling would be to show both the template (PDB accession 5ej1 or any other BcsA structure) and the modeled structure with the root-mean-square-deviation value between both structures indicated.

Thus, in this new Figure S2, the presented template and model structures, the structural elements, and residues (!) should have been labeled for the reader to see the differences between the template and the model (as stated in comments to Figure 2 above).

Why is the 18-residue cellulose fragment shown in this figure? It doesn't relate to RiGT2 chemically. It should be removed, and instead, the catalytic residues of RiGT2 should be shown to indicate, where the synthetic step of RiGT2 takes place. The residues in both zoom-in squares should be labeled.

(5) Currently, in the manuscript, there is no sequence or graphical description (designation) of catalytic motifs (DxD, xED, QxxRW) in any figure or table. There is an opportunity to do so in Figures S5, S6, or S14, where the authors should highlight those residues in sequences/alignments and refer to these motifs in legends.

(6) In general, the legends to figures showing the structural information are substandard and incorrect (Figures S2 and S11). The authors should consult a protein structural expert to help them with the correct terminology. For example, the terms “geometrical plan” and “merged” are invalid and misleading. The correct terms are “3D disposition” or “architecture” (not geometrical plan) and “superposed” (not merged) – in the latter case again, with the root-mean-square-deviation value stated between superposed structures.

The graphics of Figure 11 needs to be improved (black background removed) and show semitransparent surfaces with underlying residues in stick representations of the RxxxR motif and the PilZ domain. The numbering of residues in the RxxxR motif is not indicated.

What is RiPilZ? It is also unclear what “upper row: backside, lower row: front side” means. The authors need to consult the structural expert and use unambiguous descriptions.

(7) In Figures S14a and S14b the authors aligned selected entries but do not define, from which databases those entries were taken. These accessions need to be defined and the complete information provided in the legend to these panels. Again, there are still serious errors in the description of this legend (the amino acids-this term is incorrect as outlined in the original report!). At

least one of these alignments should include the structural template used for the generation of the structural model of RiGT2 (based on the BcsA-B complex; PDB accession 5ej1 or any other BcsA structure). Which chain was used as the template? It should have been chain A (not B) if 5ej1 was used.

(8) Could the authors provide an unrooted phylogenetic tree for Figure S13b instead of a circle tree (just like in Figure S13a)? Clearly, the identities of entries cannot be read from the images of Figures S13a and S13b. Thus, the authors need to include additional files for both trees (the list of aligned sequences and corresponding alignment files for both panels in Fasta formats).

(9) Supplementary Information is the integral component of the manuscript. The English language used throughout the text of the Supplementary Information manuscript is unsatisfactory and contains serious mistakes in both expert terms and English syntax. This needs to be checked by a structural expert (as pointed out above) and by a native English speaker. Just like it was done with the text provided in the main body of the article.

(10) Finally, a key paper was very recently published by Purushotham et al. in Scientific Advances on 11 November 2022 "Mechanism of mixed-linkage glucan biosynthesis by barley cellulose synthase-like CslF6 (1,3;1,4)- β -glucan synthase", which is highly relevant to this work. This needs to be acknowledged or at least mentioned in some form in the current manuscript.

Reviewer #3 (Remarks to the Author):

The authors have responded to all the comments that I made in the original review. They have also responded to comments from the two other reviewers, and I have no further comments or suggestions. The revised version is good.

RESPONSES TO REVIEWER'S COMMENTS

Reviewer #1:

The authors addressed this reviewer's questions thoroughly.

By testing the PilZ domain again, the authors provided new information about the role of PilZ in MLG synthesis different from cellulose synthase. All the new figures look good, but I want the authors to make fig 5 anti-MLG panel more visible. It is difficult to find the signals.

Authors' response:

Thank you for your feedback. We appreciate your positive comments on our response to the reviewer's questions. We have taken your suggestion into consideration and made the necessary adjustments to enhance the visibility of the anti-MLG panel in Figure 4.

Referee #2 comment 1:

The title of the manuscript ("The Gram-positive bacterium *Romboutsia ilealis* has a GT polysaccharide synthase that can produce mixed-linkage (1,3;1,4)- β -D-glucans") is clearer, although it can be improved. I would advise the authors against the usage of the "GT" acronym in the title, as Nature Communication is a multidisciplinary journal.

More appropriate titles could be:

"The Gram-positive bacterium *Romboutsia ilealis* has a glycosyl transferase that can produce (1,3;1,4)- β -D-glucans"

or

"The Gram-positive bacterium *Romboutsia ilealis* has a polysaccharide synthase that can produce (1,3;1,4)- β -D-glucans"

Authors' response to comment 1:

We are grateful for all the constructive feedbacks. We have changed the title to "The Gram-positive bacterium *Romboutsia ilealis* has a polysaccharide synthase that can produce (1,3;1,4)- β -D-glucans".

Referee #2 comment 2:

The term mixed-linkage (acronym MLG) and other derivatives (e.g. MLGs, MLG biosynthesis, MLG synthase) used in the manuscript are chemically incorrect and should be removed from the manuscript. The authors need to refer to the term "(1,3;1,4)- β -D-glucan" in all instances. There is a contradiction in the MLG term, as the MLG term could also indicate other mixed-linkage biopolymers such as e.g. (1,4;1,6)- α -D-glucans (branched α -glucans or amylopectins).

Authors' response to comment 2:

Thank you for bringing this to our attention. In response to your concerns, we have made replaced "MLG" by "(1,3;1,4)- β -D-glucans" throughout the manuscript.

Referee #2 comment 3:

Figure 2. The 3D structural models of RiGT2. The RiGT2 3D structural models were generated using AlphaFold2. RiGT2 consists of transmembrane (TM) region colored in blue, the cellulose synthase-like GT domains shown in pink, including three cytosolic interface helices (IF) colored in green and the C-terminal PilZ domain colored in orange. The gating loop is colored in pale yellow.

Currently, this figure is insufficient and doesn't explain the critical structural features of RiGT2. In this figure, the key catalytic residues (DxD, xED, QxxRW) illustrated in cpk stick representations, should have been shown, and labeled for the reader to see the catalytic machinery in the GT region (shown as GT-region, not "GT").

In the legend of Figure 2, the ranges of residues (or their numbering) of the TM-region, GT-region, and PilZ domain should have been included.

Lines 737-738 Why "models"? How many models were generated?

Lines 141-149 The text referring to Figure 2 is full of mistakes. The text should be consulted with the structural expert and the errors should be corrected (usage of domain, motif/motifs, polymerization of the β -glucan; containing the polymerized acceptor that incorporates incoming Glc residues). The terms and statements in brackets are incorrect.

Lines 346-347 "The secondary structure predictions were according to the structures from the AlphaFold2 predictions" (this sentence is incomplete).

Lines 737-741 The text of the legend is substandard and incomplete. It needs to be rewritten after the new Figure 2 is generated. The claim of "the gating loop" is unsubstantiated and should be removed.

Authors' response to comment 3:

Thank you for the highly relevant feedback. To address all concerns, we sought the expertise of structural biologists (VF and CD). VF and CD have revised all structural biology content including generation of models, analysis, text and figures.

Briefly, Figure 2 (now Figure 6) now includes the requested information, such as the depiction of key catalytic residues (DxD, xED, QxxRW). We have incorporated a new section titled "Structure analysis of theoretical RiGT2 models" in the text. In this section, we perform a detailed analysis of the theoretical structures generated by AlphaFold2 and SWISS-MODEL. We also analyze the differences of the channels in *CsBcsA*, *HvCsIF6*, and the RiGT2 models, as shown in Figure S12 and S13. We also analyzed the key side chains that contribute to the individual glucosyl-binding sites (Table S4).

Referee #2 comment 4:

Figure S2 AlphaFold structural models of RiGT2 show the same remarkable sequences as in BcsA.

The 3D structure generated by AlphaFold2 is shown in a gold color. The c-di-GMP and polysaccharide (cellulose) structures were obtained from the resolved crystal structure of BcsA-B complex in the PDB database (5ej1) by structural comparison in UCSF Chimera. The relative position of Trp from QxxRW and Asp from xED (orange) to the acceptor polysaccharide chain are shown in the dashed square (red) with zoom-in fields in solid

squares. The relative position of the first Arg (R517) from RxxxR and Asp and Ser from DxSxxG (Blue) to c-di-GMP are shown in the black squares.

General comments on Figure S2:

RiGT2 is not the structure. It is a “model“ or a “structural model” based on a template (in this case PDB accession 5ej1 or any other BcsA structure). This needs to be corrected in this legend and in the entire manuscript.

The 3D structure of (what protein) generated by AlphaFold2 is shown in a gold color? How does this figure relate to Figure 2? Does it contain the model of RiGT2 (it most likely does)? So, why show it again?

Specific comments on Figure S2:

The current form of this figure is unacceptable. It is unclear, from the legend description, what structure is shown here. How was the 3D model of RiGT2 constructed; a short description as to how it was done, should have been included in the legend.

The logical approach to structural modeling would be to show both the template (PDB accession 5ej1 or any other BcsA structure) and the modeled structure with the root-mean-square-deviation value between both structures indicated.

Thus, in this new Figure S2, the presented template and model structures, the structural elements, and residues (!) should have been labeled for the reader to see the differences between the template and the model (as stated in comments to Figure 2 above).

Why is the 18-residue cellulose fragment shown in this figure? It doesn't relate to RiGT2 chemically. It should be removed, and instead, the catalytic residues of RiGT2 should be shown to indicate, where the synthetic step of RiGT2 takes place. The residues in both zoom-in squares should be labeled.

Authors' response to comment 4:

As we mentioned in the previous response, all content relating to structural-biology analysis has been extensively revised. The newly generated model and template structure as well as the depiction of key catalytic residues (DxD, xED, QxxRW) of the model have been included in Figure 6. Therefore, the previous Figure S2 was removed and more structure-related figures were added in Figure S9 to S13 and Table S4.

Referee #2 comment 5:

Currently, in the manuscript, there is no sequence or graphical description (designation) of catalytic motifs (DxD, xED, QxxRW) in any figure or table. There is an opportunity to do so in Figures S5, S6, or S14, where the authors should highlight those residues in sequences/alignments and refer to these motifs in legends.

Authors' response to comment 5:

Thank you for the suggestion. We have now marked the catalytic motifs in Figure S10, which displays the structure-based sequence alignment of *RiGT2* and *CsBcsA*.

Referee #2 comment 6:

In general, the legends to figures showing the structural information are substandard and incorrect (Figures S2 and S11). The authors should consult a protein structural expert to help them with the correct terminology. For example, the terms “geometrical plan” and “merged” are invalid and misleading. The correct terms are “3D disposition” or “architecture” (not geometrical plan) and “superposed” (not merged) – in the latter case again, with the root-mean-square-deviation value stated between superposed structures.

The graphics of Figure 11 needs to be improved (black background removed) and show semitransparent surfaces with underlying residues in stick representations of the RxxxR motif and the PilZ domain. The numbering of residues in the RxxxR motif is not indicated.

What is RiPilZ? It is also unclear what “upper row: backside, lower row: front side” means. The authors need to consult the structural expert and use unambiguous descriptions.

Authors’ response to comment 6:

Thank you for your comment. We sincerely appreciate your feedback and acknowledge that the legends for Figures S2 and S11 did not meet the expected standards and contained incorrect terminology. Hence, we have removed the previous Figure S11 from the manuscript.

The c-di-GMP-binding PilZ domain in *RiGT2* shows relative low sequence identity with CsBcsA (Figure S10) but the characteristic c-di-GMP-binding motifs are present, including the RxxxR motif. We have added a figure (Figure S11) that compares the RxxxR motif in the open state of CsBcsA and *RiGT2*. The overall low sequence similarity of the PilZ domain of the two proteins in the PilZ indicate that there may be structural differences beyond the conserved motif that makes a more detailed analysis too speculative. Nonetheless, the similarity in the c-di-GMP-binding region suggests that *RiGT2* binds c-di-GMP. We would need to have c-di-GMP-binding data for the intact enzyme to evaluate and preferably also the experimental structure to investigate this further. However, as we mentioned previously, full-length *RiGT2* tends to aggregate, which presents challenges for examining substrate binding and determining the structure.

Referee #2 comment 7:

In Figures S14a and S14b the authors aligned selected entries but do not define, from which databases those entries were taken. These accessions need to be defined and the complete information provided in the legend to these panels. Again, there are still serious errors in the description of this legend (the amino acids-this term is incorrect as outlined in the original report!). At least one of these alignments should include the structural template used for the generation of the structural model of *RiGT2* (based on the BcsA-B complex; PDB accession 5ej1 or any other BcsA structure). Which chain was used as the template? It should have been chain A (not B) if 5ej1 was used.

Authors’ response to comment 7:

The protein sequence aligned in current Figure S17 were taken from NCBI database. Proteins from four species (*Clostridium nigeriense* [WP_066891591.1]; *Niameybacter massiliensis* [WP_053986099.1]; *Clostridium cuniculi* [WP_133015619.1]; *Sarcina ventriculi* [WP_195925072.1]) with highest sequence identity against *RiGT2* (WP_180703307.1) were selected as (1,3;1,4)- β -D-glucan synthase candidates. Four *Clostridium* cellulose synthases (CcsA) were selected (*Clostridium vincentii* [PRR81681.1]; *Clostridium chromiireducens*

[OPJ65957.1]; *Clostridium oryzae* [OPJ56113.1]; *Clostridioides difficile* [VIG09653.1]) according to the study published by William Scott *et al.* in 2020. This information has been added to the legend of Figure S17a.

In the case of the AlphaFold2 model, it incorporates information from various homologous templates available in the Protein Data Bank, rather than being based on a single template structure, and the database is currently dominated by bacterial cellulose synthase homologs. For producing the sequence alignment in Figure S10, we used adjusted the initial refinement by Clustal Omega using structural information from a superposition of CsBcsA and the RiGT2 AlphaFold2 model. For comparative structure analysis with CsBcsA in the manuscript, we use either 4HG6 (the original structure with an 18-unit long cellulose chain and UDP; the of Morgan *et al.* in 2013) or the structures also including c-di-GMP (4P00 and 4P02; Morgan *et al.* 2014).

Referee #2 comment 8:

Could the authors provide an unrooted phylogenetic tree for Figure S13b instead of a circle tree (just like in Figure S13a)? Clearly, the identities of entries cannot be read from the images of Figures S13a and S13b. Thus, the authors need to include additional files for both trees (the list of aligned sequences and corresponding alignment files for both panels in Fasta formats).

Authors' response to comment 8:

Thank you for the comments. The original circular tree in Figure S13b has been substituted by an unrooted phylogenetic tree and relocated in Figure S16b. The list of the aligned protein in current Figures S16a and Figure S16b are listed in Table S5 and S6.

Referee #2 comment 9:

Supplementary Information is the integral component of the manuscript. The English language used throughout the text of the Supplementary Information manuscript is unsatisfactory and contains serious mistakes in both expert terms and English syntax. This needs to be checked by a structural expert (as pointed out above) and by a native English speaker. Just like it was done with the text provided in the main body of the article.

Authors' response to comment 9:

We greatly appreciate your feedback. We want to assure you that we have taken your concerns seriously and have conducted a thorough review of the Supplementary Information manuscript. We have sought the expertise of a structural expert and a native English speaker to address any issues with terminology and English language usage.

Referee #2 comment 10:

Finally, a key paper was very recently published by Purushotham *et al.* in Scientific Advances on 11 November 2022 “Mechanism of mixed-linkage glucan biosynthesis by barley cellulose synthase-like CslF6 (1,3;1,4)- β -glucan synthase”, which is highly relevant to this work. This needs to be acknowledged or at least mentioned in some form in the current manuscript.

Authors' response to comment 10:

Thank you for providing this information. The study has been discussed in the main text (Line 356 to 360). Additionally, we also prepared another homology model of *RiGT2* based on the structure of *HvCslF6* using SWISS-MODEL. This model offers complementary information to the AlphaFold2 model, mainly regarding the channel-forming TM domain.

REVIEWERS' COMMENTS

Reviewer #2 (Remarks to the Author):

I have no further comments to the authors.

REVIEWERS' COMMENTS

Reviewer #2 (Remarks to the Author):

I have no further comments to the authors.